# Tcf1 and Lef1 provide constant supervision to mature CD8$^+$ T cell identity and function by organizing genomic architecture

Qiang Shan[1,5], Xiang Li [2,5], Xia Chen[3], Zhouhao Zeng[2], Shaoqi Zhu[2], Kexin Gai[1], Weiqun Peng [2✉] & Hai-Hui Xue [1,4✉]

T cell identity is established during thymic development, but how it is maintained in the periphery remains unknown. Here we show that ablating Tcf1 and Lef1 transcription factors in mature CD8$^+$ T cells aberrantly induces genes from non-T cell lineages. Using high-throughput chromosome-conformation-capture sequencing, we demonstrate that Tcf1/Lef1 are important for maintaining three-dimensional genome organization at multiple scales in CD8$^+$ T cells. Comprehensive network analyses coupled with genome-wide profiling of chromatin accessibility and Tcf1 occupancy show the direct impact of Tcf1/Lef1 on the T cell genome is to promote formation of extensively interconnected hubs through enforcing chromatin interaction and accessibility. The integrative mechanisms utilized by Tcf1/Lef1 underlie activation of T cell identity genes and repression of non-T lineage genes, conferring fine control of various T cell functionalities. These findings suggest that Tcf1/Lef1 control global genome organization and help form intricate chromatin-interacting hubs to facilitate promoter-enhancer/silencer contact, hence providing constant supervision of CD8$^+$ T cell identity and function.

[1] Center for Discovery and Innovation, Hackensack University Medical Center, Nutley, NJ 07110, USA. [2] Department of Physics, The George Washington University, Washington, DC 20052, USA. [3] Department of Critical Care Medicine, Fuxing Hospital, Capital Medical University, Beijing 100038, China. [4] New Jersey Veterans Affairs Health Care System, East Orange, NJ 07018, USA. [5]These authors contributed equally: Qiang Shan and Xiang Li. ✉email: wpeng@gwu.edu; haihui.xue@hmh-cdi.org

All immune cells are differentiated from multipotent hematopoietic stem cells (HSCs) following relatively well-defined maturation steps[1]. In the differentiation processes, lineage-restricted transcription factors (TFs) are induced and in turn establish identity of a specific cell type[2–4]. While cell identity is thought to be maintained by heritable epigenetic modifications of DNA and histones[5], it is suggested that the lineage-restricted TFs remain necessary to provide constant supervision of cell identity[6]. For example, while the PAX5 TF is necessary for committing early B cell progenitors to the B cell lineage[7], deletion of PAX5 in mature B cells causes their dedifferentiation to acquire HSC-like properties and even give rise to T lineage cells[8]. Additionally, ablating FOXP3 in immunosuppressive regulatory T cells induces genes characteristic of proinflammatory responses[9]. TFs, including lineage-restricted ones, frequently show pervasive occupancy in the genome[6], and their interplay with three-dimensional (3D) genome is well recognized[10]. A few TFs, such as Bcl11b, are closely associated with extensive rewiring of 3D genome during lineage differentiation from HSCs[11,12]. It is therefore of considerable interest to investigate whether the 3D genome organization underlies TF-mediated gate-keeping of cell identity. In the case of B cells, the genome architecture is perturbed by PAX5 deletion during the processes of both establishing and maintaining B cell identify[12,13]. One may infer that a lineage-defining TF exerts a genome-organizing function to supervise cell identity; however, it remains unclear whether this inference could be a generalizable concept in immune cells. In addition, there is a prevalent disconnect between changes in 3D genome and transcriptome thus far, and a causative effect of 3D genome rewiring on transcriptional output remains obscure in a given immune cell type.

CD8[+] T lymphocytes are essential for mounting protective cellular immune responses against pathogenic antigens and malignantly transformed cells[14,15]. During the early stage of T cell development in the thymus, key TFs including Tcf1, Bcl11b and Gata3 are induced to commit early thymic progenitors to the T cell lineage[2]. These TFs remain in commission to promote the production of CD4[+] or CD8[+] single positive T cells at later stages[16]. While Tcf1 and its homologue Lef1 TFs are dispensable for CD8[+] lineage choice, they are essential for establishing CD8[+] T cell identity by suppressing CD4[+] lineage-associated genes[17,18]. It is well established that the *Cd4* gene silencing in CD8[+] T cells is epigenetically maintained and heritable through subsequent mitoses[19]. It is not known if this is a general rule for all aspects of CD8[+] T cell identity after CD8[+] T cells complete intrathymic maturation and populate the peripheral lymphoid organs.

Tcf1 and Lef1 are members of the Tcf/Lef subfamily of high-mobility-group (HMG) proteins, containing a single HMG DNA-binding domain with high-degree conservation[20]. The HMG proteins are known to interact with minor groove of the DNA double-helix and thus bend DNA to modulate DNA structure[21]. Specifically, Lef1 is shown to bind a minimal TCRα enhancer and induce a sharp bend in DNA in vitro; the DNA bending, in turn, facilitates interaction of Ets and Runx family TFs that bind to sequences flanking the Lef1 site[22,23]. It has been postulated that Tcf1/Lef1 TFs may have a structural function in T cells; however, it remains unknown if such a structural process is operating in vivo and has functional impact.

In this work, we employ multiomics approaches including high-throughput chromosome-conformation-capture sequencing (Hi-C) to demonstrate that Tcf1/Lef1 TFs critically regulate global genomic organization on multiple scales in mature CD8[+] T cells. By applying a Hi-C analytical tool developed in house, we integrate Tcf1/Lef1-dependent changes in chromatin accessibility, super enhancer activity, and chromatin interactions within a genome architectural network in CD8[+] T cells, and establish links of those changes with transcriptional output. Our comprehensive analyses show that Tcf1 and Lef1 function as genome organizers and provide constant supervision of CD8[+] T cell identity and function.

## Results

**Tcf1 and Lef1 regulate genomic organization in CD8[+] T cells.** Tcf1 and Lef1 TFs are functionally redundant in establishing CD8[+] T cell identity during late stages of thymic development[17,20]. To specifically determine their functions in mature CD8[+] T cells, *hCD2-Cre*, which is most active in mature T cells in the periphery[24,25], was crossed to *Tcf7* (encoding Tcf1)- and *Lef1*-floxed alleles to generate *hCD2-Cre[+] Rosa26[GFP]Tcf7[+/+]Lef1[+/+]* and *hCD2-Cre[+]Rosa26[GFP] Tcf7[fl/fl]Lef1[fl/fl]* mice (called WT and dKO, respectively). Unlike *Cd4-Cre*-mediated Tcf1/Lef1 deletion, the use of *hCD2-Cre* did not cause derepression of CD4 coreceptor in dKO mature CD8[+] T cells (Supplementary Fig. 1a), validating specific deletion in post-thymic T cells. To directly address the postulated structural function of Tcf1/Lef1 TFs in vivo, we performed Hi-C on WT and dKO naïve CD8[+] T cells in two biological replicates (Supplementary Fig. 1b). The Hi-C libraries were reproducible between the replicates (Supplementary Fig. 1c), which were then pooled for most downstream analyses to improve sensitivity (except for identification of differential chromatin loops and interactions).

To capture Tcf1 binding events without relying on a single epitope of Tcf1 protein, recombinant full-length Tcf1 protein was used as an immunogen to generate a new Tcf1 antiserum, and its specificity and ability to immunoprecipitate Tcf1 protein were validated (Supplementary Fig. 2a–c). We then used the Tcf1 antiserum for ChIP-seq on WT or Tcf1-deficient CD8[+] T cells, which gave strong signal to noise ratio as observed at known Tcf1 target gene loci including *Cd4* and *Tcf7* itself (Fig. 1a) (the antiserum will be made available upon request). Using MACS2[26] with stringent criteria (fourfold enrichment, $p < 10^{-5}$ and FDR < 0.05), we identified 19,042 high-confidence Tcf1 binding peaks in WT CD8[+] T cells. In motif analysis with HOMER[27], we applied a position-weight matrix to determine the likelihood that a Tcf1 peak resulted from direct binding to the consensus Tcf/Lef motif. This approach provided a quantifiable "motif score" that allowed us to discern direct vs. indirect Tcf1 binding events. In total, 2,866 Tcf1 peaks with a motif score >7 were highly enriched in the Tcf/Lef motifs and were considered Tcf1 direct binding sites (Supplementary Fig. 2d), while 5,197 Tcf1 peaks with a motif score <3 lacked the Tcf/Lef motifs and were considered Tcf1 indirect binding sites (Supplementary Fig. 2d). A comparison of Motif[+] Tcf1 peaks with those with intermediate motif scores and Motif[−] ones in molecular analyses can thus help discern if direct Tcf1 binding is associated with a preferred functional output. For example, while Motif[−] Tcf1 indirect binding sites were more frequently found in gene promoters (defined as ±1 kb region flanking gene transcription start sites, TSS), Motif[+] Tcf1 direct binding sites were predominantly localized in regions distal to promoters (gene body and intergenic regions) (Supplementary Fig. 2e). Tcf1 direct binding sites harbored consensus Tcf/Lef motifs as expected, while Tcf1 indirect binding sites were highly enriched with motifs of ETS family TFs (Supplementary Fig. 2f).

On the top hierarchy of genomic organization, AB compartments are recognized as discernible large-scale structures, with A compartment being more transcriptionally active and B compartments being more silent[28]. Analysis of the compartment scores in WT CD8[+] T cells indicated that Tcf1 binding peaks were highly associated with A compartment (Fig. 1b). Comparison between WT and dKO CD8[+] T cells showed clearly discernable decrease in scores of compartments harboring multiple Tcf1 peaks upon loss of Tcf1/Lef1 TFs (Fig. 1c), suggesting that Tcf1 and

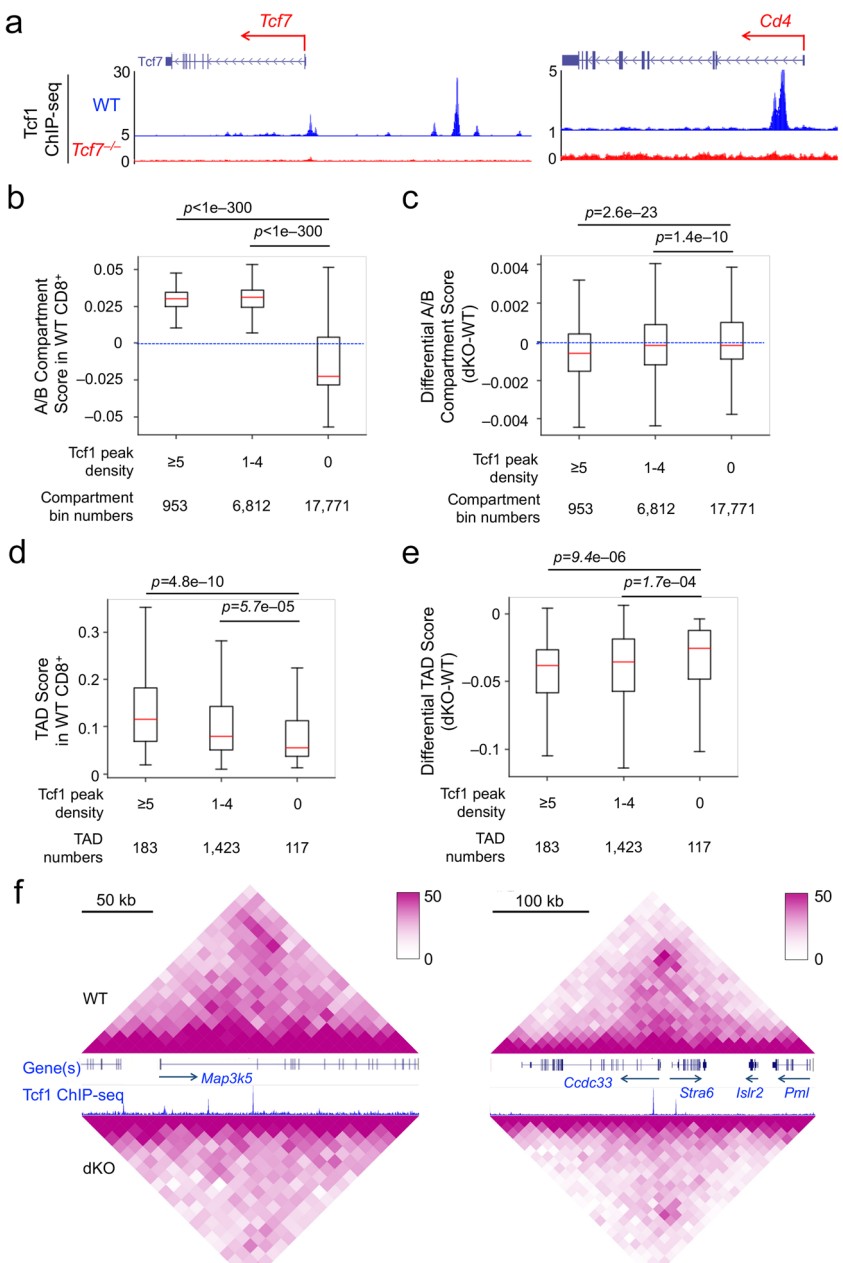

**Fig. 1 Tcf1 and Lef1 regulate large-scale genomic organization in CD8⁺ T cells. a** Tcf1 ChIP-seq tracks at the *Tcf7* and *Cd4* gene loci in WT and *Tcf7⁻/⁻* CD8⁺ T cells, with gene structure and transcription orientation displayed on top of each panel. **b** Tcf1 binding peaks are enriched in A compartments in WT CD8⁺ T cells. Compartment bins were allocated into different groups based on the density of Tcf1 peaks within each bin, and the A/B compartment scores were assessed using the Eigenvector function in the Juicer package. The blue-dotted line denotes compartment score at 0. **c** Tcf1/Lef1 deficiency diminishes compartment scores in Tcf1-bound compartments. Differential A/B compartment scores were assessed between dKO and WT CD8⁺ T cells, and then distributed into different groups based on Tcf1 peak density. The blue-dotted line denotes a non-differential score. **d** Tcf1 peaks are enriched in TADs with higher scores. TADs were identified using Arrowhead and allocated into different groups based on Tcf1 peak numbers per 100 kb in WT CD8⁺ T cells. Shown is the distribution of TAD scores in each group. **e** Tcf1/Lef1 deficiency diminishes TAD scores. Differential TAD scores were assessed between dKO and WT CD8⁺ T cells, and then distributed into different groups based on Tcf1 peak numbers per 100 kb. In (**b–e**), red line in the middle of box denotes median, box denotes interquartile range (IQR), and whiskers denote the most extreme data points that are no more than 1.5 × IQR from the edge of the box; the statistical significance was calculated using one-sided Mann–Whitney *U* test. **f** Select differential TADs detected at the *Map3k5* (left) and *Ccdc33* to *Pml* gene loci (right). Diamond graphs showing chromatin interactions in WT (top) and dKO CD8⁺ T cells (bottom) were extracted from WashU epigenome browser, with gene structures and Tcf1 ChIP-seq tracks in the middle. Genomic and color scales are displayed on the top.

Lef1 promote transcriptional activity of compartments in mature CD8⁺ T cells.

Topological associated domains (TADs) constitute the next level of genomic organization[29]. Using the Arrowhead algorithm[30], 1,723 TADs were identified in WT CD8⁺ T cells, and the TAD boundaries were enriched with CTCF, but not Tcf1 binding peaks (Supplementary Fig. 3a), based on CTCF ChIP-Seq in total T cells[31]. Analysis of TAD scores, as a measure for assessing levels of intra-TAD chromatin interactions, showed that Tcf1 peaks were associated with higher TAD scores in WT CD8⁺ T cells (Fig. 1d).

While no apparent changes in TAD boundaries were found between WT and dKO CD8[+] T cells, TAD scores were diminished by Tcf1/Lef1 deficiency, and the decrease in TAD scores in dKO CD8[+] T cells was more pronounced in TADs with higher density of Tcf1 peaks (Fig. 1e). For example, TADs covering the *Map3k5* gene and *Ccdc33* to *Pml* gene loci showed diminished intra-TAD interactions in dKO CD8[+] T cells (Fig. 1f).

To determine the impact of Tcf1/Lef1 deficiency on genomic structure on a finer scale, we analyzed chromatin interactions and loops at a 10-kb resolution. By defining interaction scores as a measure for assessing interactions of a given 10-kb bin with the rest of the chromosome, we found that the density of Tcf1 peaks was positively associated with chromatin interaction scores in WT CD8[+] T cells (Supplementary Fig. 3b). Whereas the interaction scores for chromatin regions without Tcf1 peaks were similar between WT and dKO CD8[+] T cells, those for Tcf1-associated chromatin regions were diminished in dKO CD8[+] T cells, and the decrease was more pronounced in regions with higher density of Tcf1 peaks (Supplementary Fig. 3c). To reduce noise intrinsic to Hi-C data, HiCCUPs algorithm[30] was used to determine statistically significant chromatin looping events. In WT CD8[+] T cells, we identified 11,490 high-confidence chromatin loops which encompassed enhancer–promoter (EP), promoter–promoter (PP), and enhancer–enhancer (EE) interactions, with the EP loops observed at a higher frequency of ~45%. Tcf1 peaks were highly enriched in all three groups of chromatin loops (Fig. 2a, left), and focused analysis of Motif[+] Tcf1 peaks showed that Tcf1 direct binding sites were most enriched in EE chromatin loops (Fig. 2a, right), consistent with their preferential localization in distal regulatory regions (Supplementary Fig. 2e).

Because chromatin loops rarely work in isolation[32], we examined the chromatin loops from a network perspective using the igraph platform[33] (Supplementary Fig. 3d). Densely interconnected chromatin loops formed 306 distinct hubs in WT CD8[+] T cells, and in these hubs, Tcf1 peaks were more enriched in top hub anchors with the highest connectivity, compared with bottom hub anchors with the lowest connectivity. In contrast, anchors on isolated loops were rarely bound by Tcf1 peaks (Fig. 2b). Analysis of differential chromatin loops between WT and dKO CD8[+] T cells identified 877 loops that decreased in strength significantly from WT to dKO CD8[+] T cells. On a global scale, the strength of chromatin loops was similar between WT and dKO CD8[+] T cells (Fig. 2c, rightmost column); however, the loops that harbored Tcf1 peaks at both loop anchors exhibited reduced interaction strength in dKO CD8[+] T cells, and this tendency was more evident for loops harboring Motif[+] Tcf1 peaks (Fig. 2c, leftmost two columns). Interestingly, the loops that had Motif[+] Tcf1 peak(s) in one anchor also showed weakened interaction in dKO CD8[+] T cells (Fig. 2c, middle columns). For example, a chromatin loop linking *Rbm45* and *Prkra* gene loci in WT cells, with a Tcf1 peak at the anchor harboring the *Rbm45* TSS, was diminished in strength in dKO CD8[+] T cells (Fig. 2d). On the other hand, 1,682 chromatin loops were significantly stronger in dKO than in WT CD8[+] T cells, suggesting that Tcf1/Lef1 TFs could disengage chromatin interactions, likely through indirect mechanisms (see below). However, Tcf1 peaks, especially the Motif[+] Tcf1 peaks, were more frequently associated with WT-specific chromatin loops than with dKO-specific loops, as determined with enrichment analysis (Fig. 2e). These observations suggest that Tcf1/Lef1 TFs regulate genomic organization in CD8[+] T cells across multiple scales, and promote chromatin interactions at their direct binding sites.

**Tcf1 and Lef1 coordinate ChrAcc and SE with chromatin looping.** During thymic developmental stages, Tcf1 is associated with increased chromatin accessibility (ChrAcc) for activation of

T-lineage transcriptional program[34–36]. To investigate the impact of Tcf1/Lef1 TFs on mature CD8[+] T cells, we used DNase-sequencing to map ChrAcc in WT and dKO CD8[+] T cells[11], which formed distinct clusters based on principal component analysis (PCA) (Supplementary Fig. 4a). Among a total of 28,827 ChrAcc sites, 987 sites (3.4%) were more open in WT CD8[+] T cells, while 576 sites (2%) showed significantly increase in dKO CD8[+] T cells (Fig. 3a). Stratifying with Tcf1 peaks indicated that WT-specific ChrAcc sites were predominantly bound by Tcf1 (92%, Fig. 3a), and about one third of these Tcf1 peaks were Motif[+] Tcf1 direct binding sites (Fig. 3b). In contrast, dKO-specific ChrAcc sites were less frequently bound by Tcf1 (41%, Fig. 3a), and less than 10% of these Tcf1 peaks were Motif[+] (Fig. 3b). Chromatin accessibility is more dynamic in enhancers than in promoters[37], and we therefore examined ChrAcc at Tcf1 peaks located in distal regulatory regions (>1 kb from TSS). ChrAcc at Motif[+] Tcf1 direct binding sites exhibited significant decrease in dKO compared with WT CD8[+] T cells, whereas that at Motif[−] Tcf1 indirect binding sites was similar between WT and dKO CD8[+] T cells, with Tcf1 peaks of intermediate motif scores showing modest changes (Fig. 3c). These data indicate that Tcf1/Lef1 TFs, especially at their direct binding locations, have a predominant function in maintaining a chromatin-accessible state in mature CD8[+] T cells.

We then examined the connection between chromatin accessibility and looping. WT-specific ChrAcc sites were more strongly enriched in WT-specific chromatin loops, while dKO-specific ChrAcc sites showed stronger enrichment in dKO-specific chromatin loops (Fig. 3d), indicating that chromatin looping formation is closely correlated with chromatin open state. In line with this notion, chromatin loops harboring WT-specific ChrAcc sites at single or both anchors showed progressively decreased interaction strength in dKO compared with WT CD8[+] T cells (Fig. 3e); concordantly, chromatin loops harboring dKO-specific ChrAcc sites at single or both anchors showed progressively increased interaction strength in dKO over WT CD8[+] T cells (Fig. 3e). Because over 90% WT-specific ChrAcc sites were bound by Tcf1 (Fig. 3a), the decreased strength of chromatin loops and the associated decreased ChrAcc sites could be both attributed to direct impact by Tcf1/Lef1 TFs, highlighting their critical functions in coordinating chromatin interaction and accessibility in mature CD8[+] T cells.

Super enhancers (SEs) contribute to control of cell identity, and in T lineage cells, *Tcf7* gene locus is regulated by an SE[38,39]. We performed H3K27ac ChIP-seq on WT and dKO CD8[+] T cells, which exhibited distinct profiles on PCA (Supplementary Fig. 4b). By use of the ROSE algorithm[39] to stitch and rank H3K27ac-enriched regions called by SICER[40], 1,160 SEs were identified in WT CD8[+] T cells, and these SEs were enriched with Tcf1 binding peaks (Fig. 3f). By comparing SEs identified in WT and dKO CD8[+] T cells with edgeR at FDR < 0.05, we found that 174 and 163 SEs exhibited consistently stronger H3K27ac signals in WT and dKO CD8[+] T cells, respectively. Because of the larger scale of SEs where not all individual enhancers exhibited similar changes, these 'differential' SEs were hereby referred to as WT- or dKO-prepotent SEs rather than WT- or dKO-specific SEs CD8[+] T cells. While Tcf1 peaks were enriched in both WT- and dKO-prepotent SEs, Tcf1 binding events were more enriched in WT-prepotent SEs (Fig. 3g). In examining the connection of SEs with chromatin looping, we observed that chromatin loops that overlapped with WT-prepotent SEs at single or both anchors showed progressively decreased interaction strength in dKO compared with WT CD8[+] T cells (Fig. 3h). On the other hand, chromatin loops that overlapped with dKO-prepotent SEs at both anchors, but not at a single anchor, showed significantly increased interaction strength in dKO over WT CD8[+] T cells (Fig. 3h). These observations suggest that Tcf1/Lef1 TFs concordantly

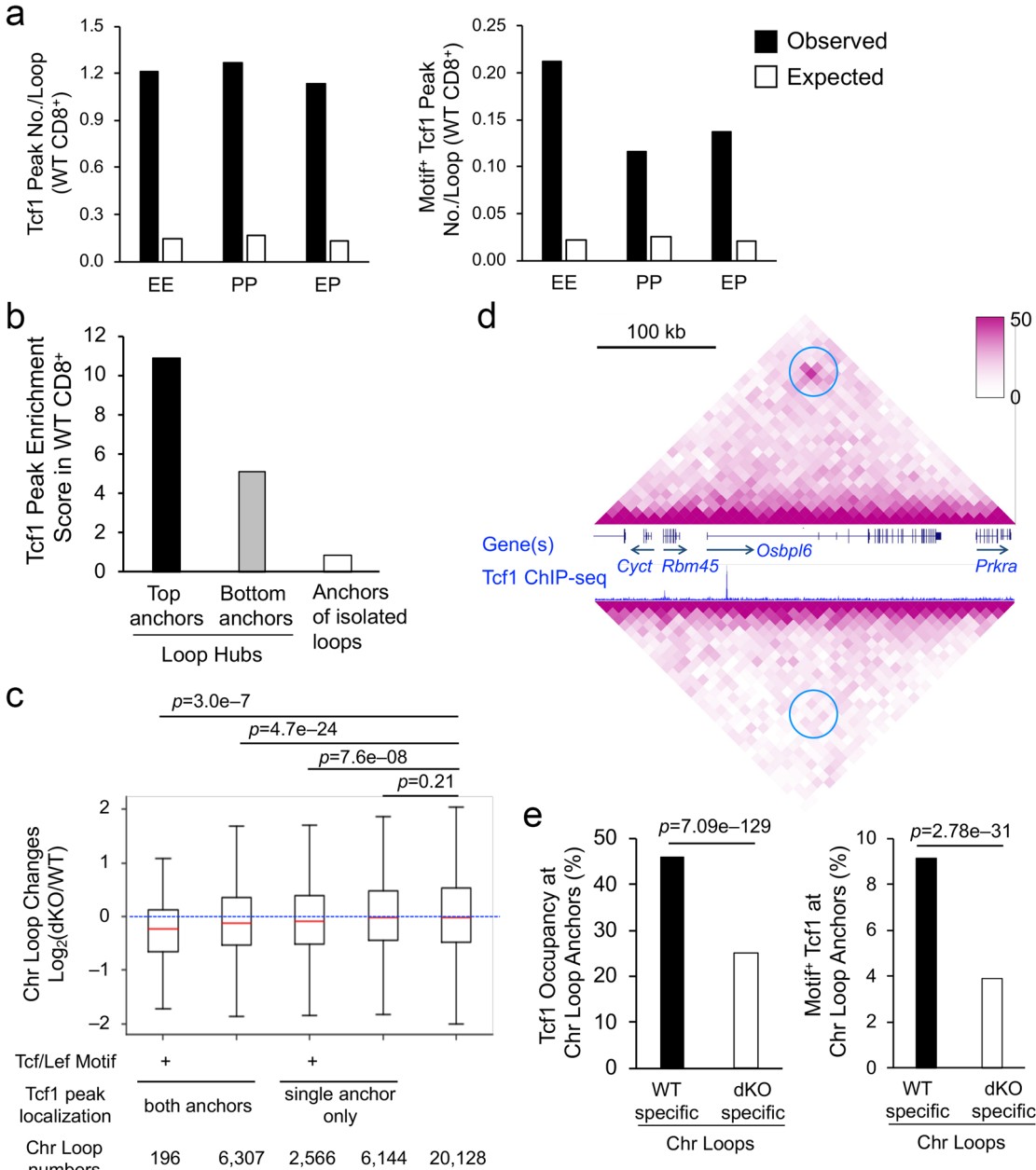

**Fig. 2 Tcf1 and Lef1 control interaction strength in chromatin loops in CD8+ T cells. a** Tcf1 peaks are enriched in anchors of chromatin loops. Chromatin loops were called using HiCCUPS and allocated into different groups according to anchor annotation in promoters (P) or enhancers (E). The observed numbers of all (left) and Motif+ (right) Tcf1 peaks per loop were identified for each group, and the expected numbers were based on genome average. **b** Tcf1 peaks are enriched in the hubs of the chromatin loop network. From the network of chromatin loops in WT CD8+ T cells, 306 hubs were identified as highly interconnected clusters of interactions. Anchors of hubs were ranked according to interaction frequency, and the top (No. 1) and bottom (last) anchors from each hub were collected and analyzed for Tcf1 peak enrichment. Anchors of the isolated loops were analyzed as a control. **c** Tcf1/Lef1 deficiency diminishes the strength of Tcf1-associated chromatin loops. Chromatin loops identified in WT and dKO CD8+ T cells were allocated into different groups based on the presence of total and Motif+ Tcf1 peaks at a single or both loop anchors, and changes in loop strength were assessed in each group. The blue-dotted line denotes no change in loop strength. Boxplot annotation is the same as Fig. 1e. **d** Select differential chromatin loop at the *Rbm45* to *Prkra* gene loci, as denoted by blue circles. **e** Tcf1 peaks are enriched in anchors of WT-specific chromatin loops. Differential chromatin loops were called using edgeR, and the frequency of all (left) and Motif+ Tcf1 peaks (right) at anchors of WT- or dKO-specific chromatin loops is shown. The statistical significance is assessed using one-sided binomial test.

modulate SE activity and chromatin loop strength in mature CD8+ T cells.

**Tcf1 and Lef1 repress non-T lineage-enriched genes in CD8+ T cells**. To define the impact of Tcf1/Lef1 deficiency on CD8+ T cell biology, RNA-seq analysis was performed on WT and dKO

CD8+ T cells, which showed distinct expression clusters on PCA (Supplementary Fig. 4c). By criteria of ≥2-fold expression changes and FDR < 0.05, we found that 258 genes were downregulated, while 313 genes were upregulated in dKO CD8+ T cells. Functional annotation using DAVID Bioinformatics Resources[41] showed that "immunity", "adaptive immunity" and "innate immunity" were the top enriched functional clusters in both down- and upregulated

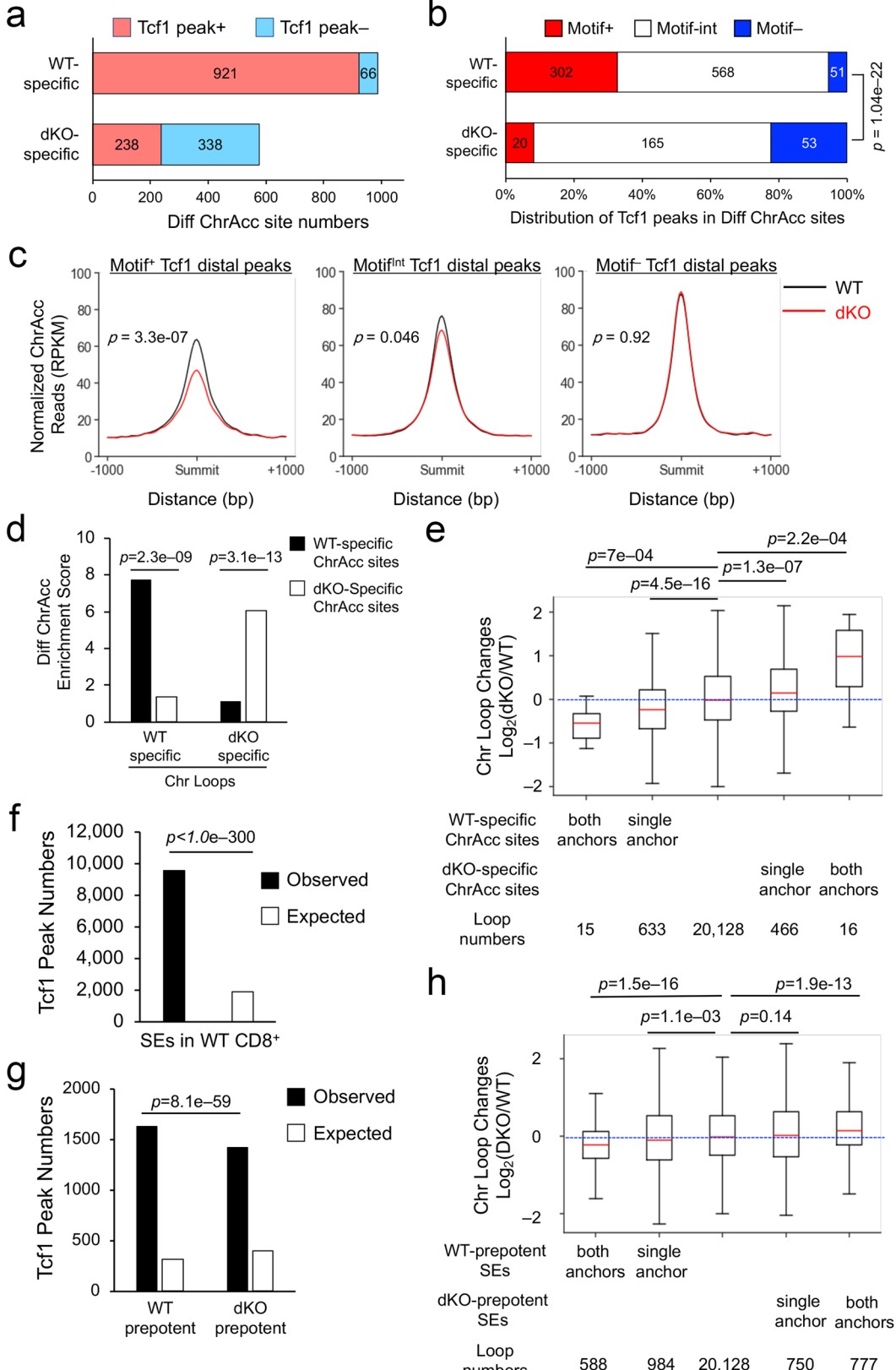

genes (Fig. 4a), suggesting a close link of Tcf1/Lef1 TFs with immune cell identity-related genes.

To further develop this notion, we retrieved RNA-Seq data from a recent study that characterized the transcriptomes of 86 immune cell populations encompassing lymphoid and myeloid hematopoietic lineages in mice[42]. We extracted lineage-enriched genes (LEGs) from

seven cell types, which included pan-T cells (including naïve CD8[+], conventional naïve CD4[+], Treg, and γδ-T cells), B cells, pan-NK (including CD27[+]CD11b[+], CD27[+]CD11b[−], and CD27[−]CD11b[+] subsets), conventional DCs (cDCs including CD4[+] and CD8[+] subsets), plasmacytoid DCs (pDCs), monocytes (including Ly6C[+] and Ly6C[−] subsets), and bone marrow-derived granulocytes (Fig. 4b,

**Fig. 3 Tcf1/Lef1-dependent changes in chromatin accessibility and super enhancer activity are concordant with changes in chromatin loop strength.** **a** Tcf1 peaks are frequently associated with WT-specific ChrAcc sites. Differential (Diff) ChrAcc sites in WT and dKO CD8$^+$ T cells were stratified with Tcf1 peaks. Values denote ChrAcc site numbers. **b** Tcf1 direct binding sites are preferably associated with WT-specific ChrAcc sites. Tcf1-bound differential ChrAcc sites were divided to Motif$^+$, Motif$^{Int}$ and Motif$^-$ groups based on strong, intermediate and weak enrichment of Tcf/Lef motif, and their distribution among WT- or dKO-specific sites was plotted, with statistical significance assessed using Chi-squared test. Values denote numbers of ChrAcc sites. **c** Tcf1/Lef1 regulate ChrAcc at direct binding sites. Motif$^+$, Motif$^{Int}$, and Motif$^-$ Tcf1 peaks in distal regions were analyzed for aggregated ChrAcc profiles in WT and dKO CD8$^+$ T cells, with statistical significance assessed using one-sided Wilcoxon signed rank test. **d** Differential chromatin loop anchors are enriched with concordant ChrAcc changes. Enrichment scores of differential ChrAcc sites were assessed at anchors of WT- or dKO-specific chromatin loops, with statistical significance calculated using one-sided Fisher's exact test. **e** Chromatin loop strength is correlated with ChrAcc state at loop anchor(s). The relative chromatin loop strength (WT/dKO) was assessed in different groups based on ChrAcc state at one or both loop anchors. **f** Tcf1 peaks are enriched in super enhancers (SEs), with statistical significance calculated using one-sided binomial test. **g** Tcf1 peaks are more enriched in WT- over dKO-prepotent SEs, with statistical significance calculated using one-sided Poisson test. The expected numbers of Tcf1 peaks in (**f**) and (**g**) were estimated based on genome average. **h** Chromatin loop strength is correlated with SE activity at loop anchor(s). The relative chromatin loop strength (WT/dKO) was assessed in different groups based on differential SE activity at one or both loop anchors. In (**e**) and (**h**), boxplot annotation is the same as Fig. 1e, statistical significance of indicated pairwise comparisons assessed using one-sided Mann–Whitney $U$ test, and the blue-dotted lines denote no change in loop strength.

and Supplementary Data 1). For a LEG, its enrichment in a specific lineage was defined as being expressed at least 5-fold higher than five out of six other lineages, with an FDR < 0.01. This definition was based on the consideration that many genes are expressed in more than one lineages during lineage development, activation, and/or differentiation process in immune responses. We then used gene set enrichment analysis (GSEA) to assess the behavior of the immune LEGs as gene sets without preset threshold. Pan-T cell LEGs were enriched in WT CD8$^+$ cells (Fig. 4c); for example, *Ccr7*, encoding the CCR7 chemokine receptor that directs naïve T cell trafficking to secondary lymphoid organ, showed diminished expression in dKO CD8$^+$ T cells (Supplementary Fig. 5a). Surprisingly, all non-T cell LEGs (including B, NK, cDC, pDC, monocytes, and granulocytes) were strongly enriched in dKO CD8$^+$ T cells (Fig. 4d). These genes include *Cd19*, *Cd79a*, and *Cd79b* that encode the Ig-α and Ig-β BCR components, *Ccl5* that augments NK activity[43], *Flt3* that is critical for homeostatic DC division[44], and *Cebpd* that encodes C/EBPδ and modulates monocyte differentiation[45] (Supplementary Fig. 5a). These observations suggest that in mature CD8$^+$ T cells, Tcf1/Lef1 TFs are required to provide constant supervision of CD8$^+$ T cell identity, to actively suppress aberrant transcription of non-T lineage-associated genes.

Within the T cell lineage, we previously reported that during thymic development, Tcf1 and Lef1 repress CD4$^+$ lineage-associated genes and effector CD8$^+$ T cell genes in thymic CD8$^+$ T cells[17]. To discern if these requirements persist in mature CD8$^+$ T cells in the periphery, we performed clustering analysis within all T cell subsets and found that effector CD8$^+$ T cells had a distinct transcriptome profile from other naive T cell subsets (Supplementary Fig. 6a). Focused analysis of naïve T cells identified naïve CD8$^+$, CD4$^+$, Treg and γδT cell LEGs (Supplementary Fig. 6b and Supplementary Data 2). GSEA showed that not only effector CD8$^+$ but also Treg and γδT cell LEGs were enriched in dKO CD8$^+$ T cells (Supplementary Fig. 6c). For example, *Gzmb* (encoding granzyme B, a key cytotoxic molecule), *Prdm1* (encoding the Blimp1 TF that promote effector T cell differentiation), *Foxp3* (the lineage-defining TF of Treg cells), and *Nrp1* (encoding a Treg surface marker Neurophilin-1)[46] were highly expressed in dKO CD8$^+$ T cells (Supplementary Fig. 5b). In contrast, both naïve CD8$^+$ and naïve CD4$^+$ T cell LEGs, which were more limited in numbers, showed modest enrichment in WT CD8$^+$ T cells (Supplementary Fig. 6d). This focused analysis on T-lineage cells indicates that Tcf1/Lef1 TFs remained necessary to suppress aberrant expression of cytotoxic effector-associated and Treg lineage-enriched genes. It should be noted, however, the increased transcripts of non-T or non-cytotoxic lineage genes in dKO CD8$^+$ T cells do not mean that Tcf1/Lef1-deficient CD8$^+$ T cells were

reprogrammed into other cell types such as B cells, DCs, or Treg cells, because they retained the capacity of inducing cytotoxic cytokines upon activation (detailed below).

**Tcf1 and Lef1 utilize ChrAcc sites as enhancers or silencers.** To determine how the impact of Tcf1/Lef1 TFs on chromatin and genomic organization is connected to transcriptional output in mature CD8$^+$ T cells, we took a gene-centric approach by pooling DEGs defined by the preset thresholds and differentially expressed lineage-enriched genes (DLEGs) defined by leading edges based on GSEA (Supplementary Data 3). The DEG + DLEG approach identified 847 up- and 283 downregulated genes in dKO CD8$^+$ T cells, which were defined as Tcf1/Lef1-repressed and -activated genes, respectively. To link differential ChrAcc with these genes, we adopted the following association rules to include ChrAcc sites (1) at the promoter region, (2) within gene body and 50 kb upstream of TSS, and (3) connected to gene promoters via chromatin loops (but not limited to differential loops between WT and dKO CD8$^+$ T cells). One hundred and seventy-five Tcf1/Lef1-repressed genes were linked to 220 differential (Diff) ChrAcc sites, with 60% showing increased ChrAcc, and the rest showing decreased ChrAcc (clusters 1 and 2 in Fig. 5a, respectively; Supplementary Data 4). Thirteen Diff ChrAcc sites in cluster 1 were at the promoters including *Gzma* and *Gzmb* (effector CD8$^+$ LEGs, Fig. 5b), and others were at distal regulatory regions, such as introns of *Blk* (a B cell LEG) and *Ctla4* (a Treg cell LEG) (Fig. 5c). Counter-intuitively, Cluster 2 Diff ChrAcc sites showed direct linkage of decreased ChrAcc with increased gene expression. These sites were found at downstream of *Fasl* (an effector CD8$^+$ LEG), introns of *Cd40lg* (a CD4$^+$ LEG), *Pax5*, and *Syk* (B cell LEGs) (Fig. 5d), upstream of *Gzmb* (Fig. 5b) and *Prdm1* (effector CD8$^+$ LEGs, Fig. 6b). This Diff ChrAcc cluster showing changes discordant with expression of linked genes may function as transcriptional silencers/repressors (see below).

On the other hand, 80 Tcf1/Lef1-activated genes were linked to 113 Diff ChrAcc sites, with most of these sites showing decrease in ChrAcc in dKO CD8$^+$ T cells (C4 in Fig. 5a; Supplementary Data 4). Fourteen genes including *Gria3* (a naïve CD8$^+$ LEG, encoding glutamate receptor C) and *Pou2af1* (an OCT TF-associated transcriptional coactivator) exhibited decreased ChrAcc at their promoters (Fig. 5e). Other genes had decreased ChrAcc at distal regulatory regions, such as two upstream sites at *Tubb3* (tubulin β3) and an intron site in *Bach2* (Fig. 5f). The minor C3 cluster showed increased ChrAcc in dKO CD8$^+$ T cells, discordant with gene expression, and might have repressive regulatory functions.

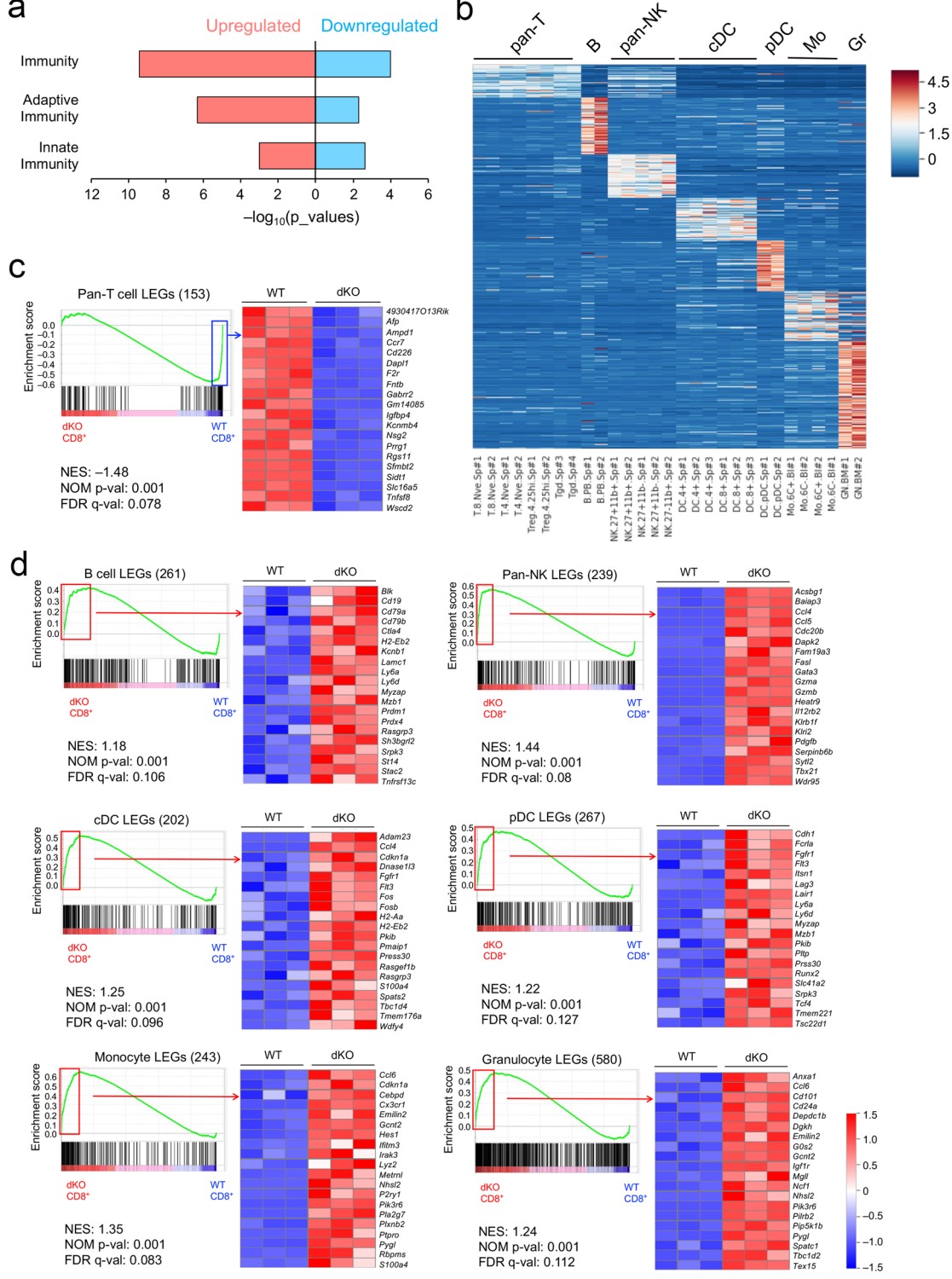

**Fig. 4 Tcf1 and Lef1 are required for repressing non-T lineage-enriched genes in mature CD8+ T cells. a** Functional annotation of up- and downregulated genes in dKO CD8+ T cells with the DAVID Bioinformatics Resources. The most enriched categories and corresponding *p* values (output from DAVID) are shown. **b** Heatmap showing lineage-enriched genes (LEGs) for pan-T, B, pan-NK cells, conventional DC (cDC), plasmacytoid DC (pDC), monocytes (Mo) and bone marrow granulocytes (Gr) based on transcriptomic analysis of multiple immune cell lineages (GSE109125). Color scale represents z-score transformed from expression data. **c** pan-T cell LEGs are enriched in WT over dKO CD8+ T cells, where 61 of 153 genes are at the leading edge. **d** non-CD8+ T cell LEGs are enriched in dKO over WT CD8+ T cells, where 89 of 261 B cell LEGs, 83 of 239 pan-NK cell LEGs, 77 of 202 cDC LEGs, 79 of 267 pDC LEGs, 107 of 243 monocyte LEGs, and 167 of 580 granulocyte LEGs are at the leading edge. For all GSEA, enrichment plot for each gene set is shown, and also marked are NES (normalized enrichment score), NOM *p* val (nominal *p* values), and FDR q-val (false discovery rate q values) as output from GSEA. Red and blue rectangles mark genes in the leading edge, enriched in WT and dKO CD8+ T cells, respectively. Top enriched genes in each set are shown in heatmaps.

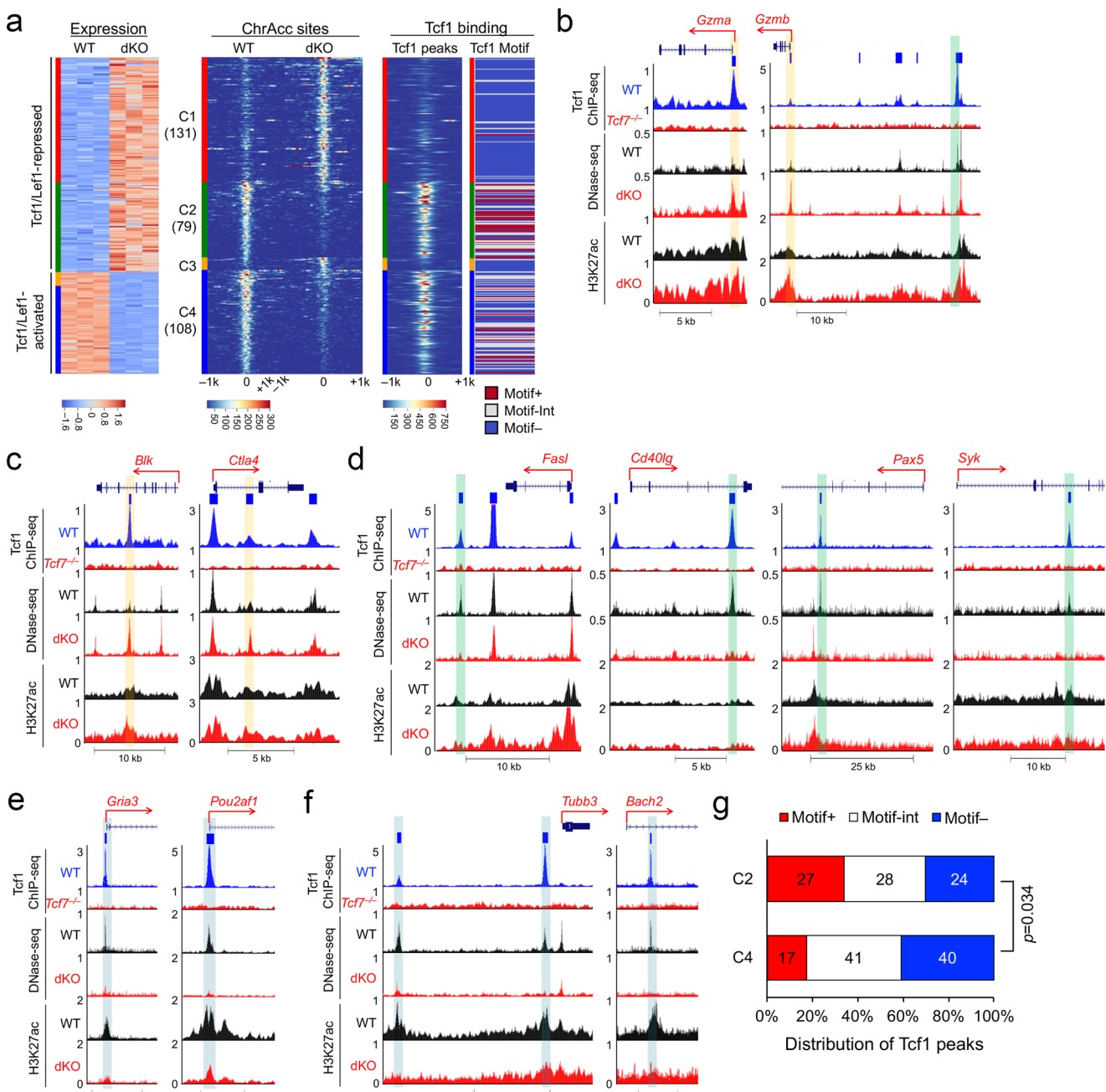

**Fig. 5 Tcf1 and Lef1 control chromatin accessibility for target gene regulation. a** Differential ChrAcc site-associated Tcf1/Lef1 target genes were displayed as Tcf1/Lef1-repressed and -activated groups in an expression heatmap (left). In each group of genes, the associated differential ChrAcc sites were clustered according to the changes in ChrAcc (middle panel). Values in parentheses denote numbers of Diff ChrAcc sites in each cluster. In the right panels, the left column displays the signal strength of Tcf1 peaks at each ChrAcc site, and the right column displays Tcf/Lef motif enrichment category for each site as denoted with a horizontal line of distinct color code. **b**–**f** At select gene loci, shown are Tcf1 ChIP-seq tracks in WT and *Tcf7−/−* CD8+ T cells (top), DNase-seq tracks (middle) and H3K27ac ChIP-seq tracks (bottom) in WT and dKO CD8+ T cells. Whole or partial gene structure and transcription orientation are displayed on top of each panel. Blue horizontal bars denote MACS2-called high-confidence Tcf1 binding peaks. **b**, **c** Tcf1/Lef1-repressed genes with increased ChrAcc sites in dKO CD8+ T cells, at promoter regions of *Gzma* and *Gzmb* (**b**) or distal regulatory regions of *Blk* and *Ctla4* (**c**), as marked by yellow vertical bars. **d** Tcf1/Lef1-repressed genes with decreased ChrAcc sites at distal regulatory regions of *Fasl, Cd40lg, Pax5* and *Syk* in dKO CD8+ T cells, as marked by green vertical bars. A distal site upstream of *Gzmb* in (**b**) belongs to this category. **e**, **f** Tcf1/Lef1-activated genes with decreased ChrAcc sites at promoter regions of *Gria3* and *Pou2af1* (**e**) or distal regulatory regions of *Tubb3* and *Bach2* (**f**), in dKO CD8+ T cells, as marked by blue vertical bars. **g** Distribution of Tcf1 peaks with different motif enrichment (Motif+, Motif-Int and Motif−) in Tcf1-bound Diff ChrAcc sites in C2 and C4 clusters as defined in (**a**). Values denote actual numbers of Tcf1-bound ChrAcc sites in each motif enrichment group. The statistical significance is assessed using the Chi-squared test.

Parsing Diff ChrAcc clusters with Tcf1 occupancy showed that Tcf1 peaks were predominantly enriched in C2 and C4 clusters where ChrAcc was diminished in dKO CD8+ T cells (Fig. 5a). Of note, high-confidence Tcf1 peak(s) were found to overlap with all 79 ChrAcc sites in C2 cluster and 98 out of 108 sites in C4 cluster, and these Tcf1-bound ChrAcc sites exhibited a range of motif scores for Tcf/Lef motif (Fig. 5a, right panels). This observation suggests that Tcf1/Lef1 TFs have a direct function in keeping

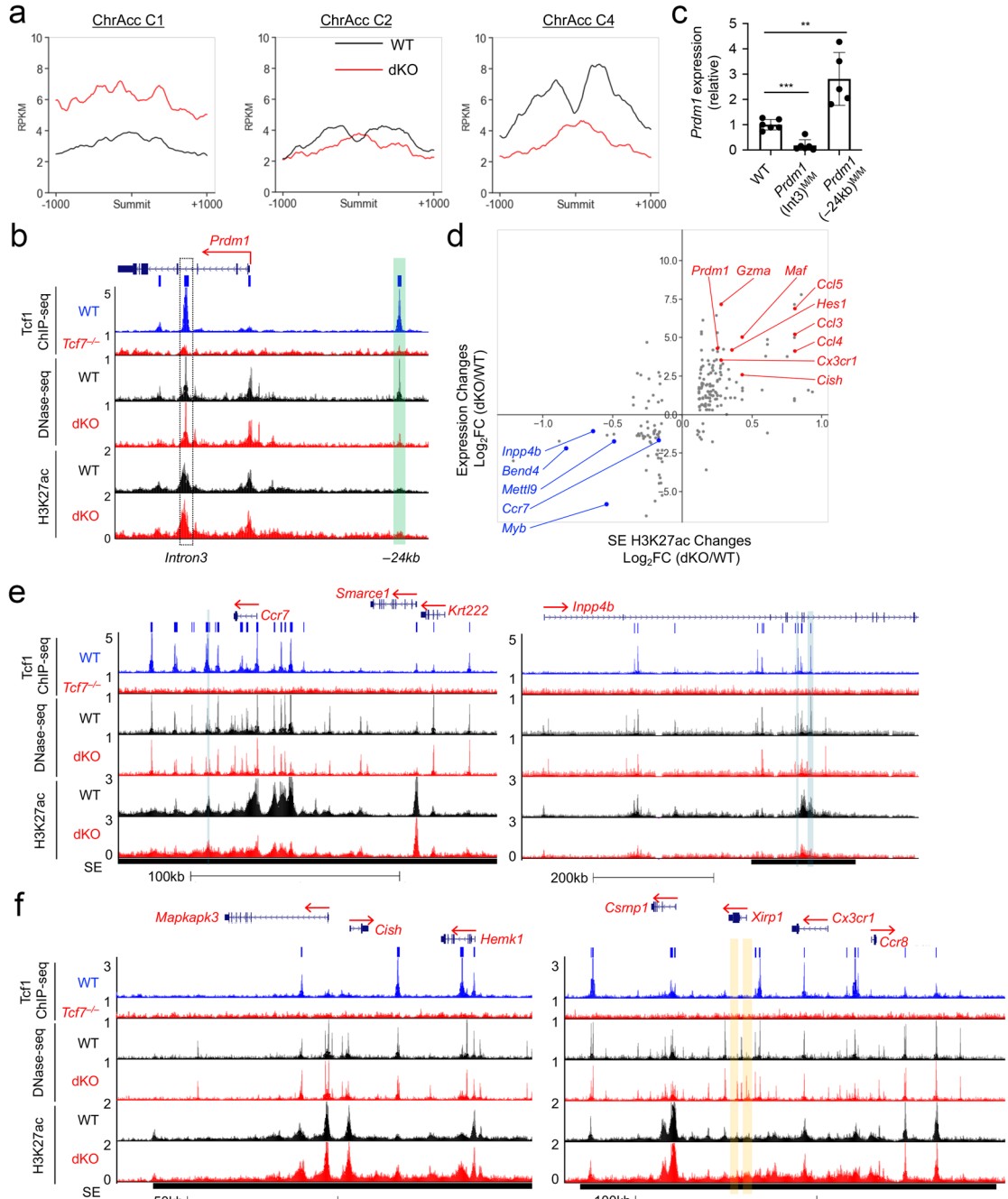

**Fig. 6 Tcf1 and Lef1 modulate silencer and super enhancer activity for target gene regulation. a** H3K27ac profiles at differential ChrAcc clusters C1, C2 and C4 from Fig. 5a. **b** Tcf1 ChIP-seq, DNase-seq and H3K27ac ChIP-seq tracks at the *Prdm1* locus. The 24 kb upstream Tcf1-bound ChrAcc site is in the C2 cluster, as marked with a green vertical bar. The intron 3 Tcf1-bound ChrAcc site, marked with dotted rectangle, was similar in ChrAcc signal strength in WT and dKO CD8$^+$ T cells. **c** *Prdm1* expression was measured with RT-PCR in WT, *Prdm1*(Int3)$^{M/M}$ and *Prdm1*(−24 kb)$^{M/M}$ naïve CD8$^+$ T cells. Data are means ±S.D. from ≥3 independent experiments ($n ≥ 5$). **, $p < 0.01$; ***, $p < 0.001$ as assessed with two-tailed Student's *t*-test for indicated pairwise comparisons. **d** Scatterplot showing correlation between H3K27ac changes on differential SEs and expression changes of SE-associated genes, with selected genes marked. **e**, **f** Tcf1 ChIP-seq, DNase-seq and H3K27ac ChIP-seq tracks at select SE-associated, Tcf1/Lef1-activated (**e**) or -repressed genes (**f**). Black horizontal bars under each panel denote SE-encompassed regions. Blue vertical bars denote C4 Diff ChrAcc sites in WT-prepotent SEs (**e**), and yellow bars denote C1 Diff ChrAcc sites in dKO-prepotent SEs (**f**).

chromatin at an accessible/open state in mature CD8$^+$ T cells. In contrast, ChrAcc sites in C1 cluster that were increased in dKO CD8$^+$ T cells showed sparse overlap with high-confidence Tcf1 peaks, indicative of indirect mechanisms by which Tcf1/Lef1 TFs restrain ChrAcc at these locations. Because C2 and C4 ChrAcc clusters were associated with opposite gene expression patterns in dKO CD8$^+$ T cells, we examined relative enrichment of Tcf1

direct binding sites within the two clusters but found only modest bias of Motif$^+$ Tcf1 peaks in the C2 cluster (Fig. 5g). We next checked H3K27ac state in the ChrAcc clusters. Both C1 and C4 ChrAcc sites showed concordant changes in H3K27ac signals, with C4 sites showing stronger H3K27ac in WT CD8$^+$ T cells than in dKO cells (Fig. 6a). Specifically, the ChrAcc sites observed in *Tubb3* upstream and *Bach2* intron regions were associated with

stronger H3K27ac signals in WT CD8$^+$ T cells, and both ChrAcc and H3K27ac signals were markedly reduced in dKO CD8$^+$ T cells (Fig. 5f). These observations are consistent with the notion that C4 ChrAcc sites are Tcf1/Lef1-dependent transcriptional enhancers to activate the expression of linked genes.

In contrast, H3K27ac signals in C2 ChrAcc clusters were substantially weaker than those in C4 clusters in WT CD8$^+$ T cells, but were similar between WT and dKO CD8$^+$ T cells (Fig. 6a, middle panel). This distinct pattern was consistently observed at the ChrAcc sites in introns of *Cd40lg*, *Pax5*, and *Syk* (Fig. 5d) and upstream of *Prdm1* (Fig. 6b). The lack of H3K27ac at C2 ChrAcc sites suggests that these chromatin-accessible sites do not function as enhancers, but may instead act as Tcf1/Lef1-dependent transcriptional silencers/repressors. To further investigate this notion, we focused on the *Prdm1* gene, which was upregulated in dKO CD8$^+$ T cells. The *Prdm1* locus contained at least two high-confidence Tcf1 peaks, one at the −24 kb upstream of *Prdm1* TSS and the other in intron 3, and both Tcf1 binding sites exhibited strong chromatin accessibility in WT CD8$^+$ T cells (Fig. 6b). The *Prdm1* −24 kb ChrAcc site was in C2 and became less accessible in dKO CD8$^+$ T cells, and H3K27ac signals at this site were at the background level in both WT and dKO CD8$^+$ T cells (Fig. 6b). In contrast, the *Prdm1* intron3 ChrAcc site in WT CD8$^+$ T cells remained similarly accessible in dKO CD8$^+$ T cells. This site showed strong H3K27ac signals in WT CD8$^+$ T cells, which were modestly elevated, if any, in dKO CD8$^+$ T cells (Fig. 6b). We have previous targeted each site in germline using CRISPR/Cas9 approach and generated *Prdm1*(−24 kb)$^M$ and *Prdm1*(Int3)$^M$ alleles separately[47,48]. Interestingly, *Prdm1*(−24 kb)$^{M/M}$ naïve CD8$^+$ T cells showed elevated *Prdm1* expression than WT cells (Fig. 6c), supporting the notion that the *Prdm1* −24 kb site is a Tcf1/Lef1-dependent transcriptional repressor element. In contrast, *Prdm1* expression was reduced in *Prdm1*(Int3)$^{M/M}$ compared with WT CD8$^+$ T cells (Fig. 6c), suggesting the *Prdm1* intron 3 site functions as a Tcf1/Lef1-independent transcriptional enhancer to support basal *Prdm1* transcription in naïve CD8$^+$ T cells.

**Tcf1 and Lef1 modulate super enhancer activity for gene regulation.** We next examined the contribution of Tcf1/Lef1-modulated SEs to target gene regulation. Parsing differential SEs with Tcf1/Lef1 target genes showed that changes in SE signal strength were mostly concordant with those in expression of SE-associated genes (Fig. 6d). Forty-seven Tcf1/Lef1-activated genes were associated with 35 WT-prepotent SEs (Fig. 6d, quadrant iii; Supplementary Fig. 7a, left; Supplementary Data 5). For example, *Ccr7* was within an SE that had stronger H3K27ac signals in WT than dKO CD8$^+$ T cells (Fig. 6e). Another pan-T cell LEG *Inpp4b* encompasses 785 kb, and its gene body harbored an SE with stronger H3K27ac in WT CD8$^+$ T cells (Fig. 6e). Both genes contained multiple Tcf1 peaks within the gene body, and *Ccr7* was associated with several Tcf1 peaks in its flanking genomic regions. It is of note that only one Tcf1 peak in *Ccr7* and two Tcf1 peaks in *Inpp4b* SEs overlapped with decreased ChrAcc in dKO CD8$^+$ T cells; in contrast, the SE regions extensively bound by Tcf1 showed broadly stronger H3K27ac in WT over dKO CD8$^+$ T cells (Fig. 6e). These observations suggest that Tcf1/Lef1 TFs directly modulate SE activity to activate their target gene transcription, in addition to maintaining chromatin open state at select regulatory elements.

On the flip side, 78 Tcf1/Lef1-repressed genes were associated with 62 dKO-prepotent SEs (Fig. 6d, quadrant i; Supplementary Fig. 7a, right; Supplementary Data 5). For example, *Cish* and *Cx3cr1* (effector CD8$^+$ and Treg LEGs, respectively) were linked to SEs with elevated H3K27ac in dKO CD8$^+$ T cells (Fig. 6f), and both SEs harbored multiple Tcf1 peaks in WT CD8$^+$ T cells.

While *Cish*-linked SE did not show ChrAcc changes, *Cx3cr1*-linked SE contained two increased ChrAcc sites in dKO CD8$^+$ T cells, which did not overlap with Tcf1 peaks (Fig. 6f), consistent with a pattern observed in ChrAcc C1 cluster (Fig. 5a). These observations suggest that Tcf1/Lef1 TFs are involved in dampening the activity of a group of SEs in mature CD8$^+$ T cells, but largely through indirect mechanisms.

**Tcf1 and Lef1 modulate chromatin interactions for gene regulation.** Whereas Tcf1/Lef1 deficiency affected the strength of select chromatin loops defined by stringent statistical significance, such as the one observed between *Rbm45* and *Prkra* gene loci (Fig. 2d), a more frequently observed impact was diminished chromatin interactions within a TAD or sub-TAD, as observed in the *Map3k5* locus (Fig. 1f, left panel). To systematically identify such regions and assess their effect on the expression of target genes, we adopted the approach of network analysis. We first extracted all interactions with the same directional changes between WT and dKO CD8$^+$ T cells, and then identified 3D chromatin interaction clusters based on connectivity among bins anchoring these interactions (Fig. 7a). The 3D clusters were then projected onto one-dimensional genomic regions, followed by integrative comparative analysis of numbers, strength and statistical significance of chromatin interactions among the regions. This approach identified 221 WT-specific and 194 dKO-specific chromatin interaction hubs (Fig. 7a). For example, in the interaction network of a chromosome 10 segment, a cluster harboring the *Myb* gene formed a WT-specific hub, where chromatin interactions were stronger in WT than dKO CD8$^+$ T cells (Fig. 7b). Similarly, in the network of a chromosome 11 segment, the cluster harboring the *Ccl3*, *Ccl4*, *Ccl9*, and *Ccl5* gene loci formed a dKO-specific hub (Fig. 7c). Notably, both WT- and dKO-specific hubs were enriched with Tcf1 peaks, and the Tcf1 peak enrichment was substantially higher in WT-specific hubs (Fig. 7d), indicating a predominant function of Tcf1/Lef1 TFs in promoting chromatin interactions and hub formation.

To connect the hubs to gene expression changes, hubs containing at least one DEG + DLEG promoter were identified and plotted against gene expression changes (Fig. 7e, Supplementary Data 6). This approach showed a strong association of WT-specific hubs with Tcf1/Lef1-activated genes including the MYB TF and IL-6 receptor α chain (encoded by *Il6ra*) along with several pan-T cell LEGs such as *Cpm*, *Dapl1*, *Gabrr2*, *Inpp4b*, *Prrg1,* and *Rgs11* (Fig. 7e, quadrant iii). Consistent with localization of the *Myb* gene locus within a WT-specific hub (Fig. 7b), the *Myb* promoter showed extensive interactions with its upstream regions that spanned over 100 kb, and the chromatin interactions were greatly diminished in dKO compared with WT CD8$^+$ T cells (Fig. 7f). In addition, through hub analysis, the *Myb* promoter was linked to three ChrAcc sites showing significant reduction in dKO CD8$^+$ T cells (Fig. 7b, g), complementing the empirical linking rule because these sites were >50 kb upstream of Myb TSS. Furthermore, the *Myb* locus was in a WT-prepotent SE (Figs. 6d, 7g). Similarly, the *Inpp4b* (a pan-T LEG) locus was in a WT-specific hub (Supplementary Fig. 7b), showing decreased intronic interactions in dKO CD8$^+$ T cells (Supplementary Fig. 7c). The *Inpp4b* promoter was linked to two ChrAcc sites showing reduction in dKO CD8$^+$ T cells (Supplementary Fig. 7b, Fig. 6e), and the locus harbored a WT-prepotent SE (Fig. 6d, e). These analyses indicated that Tcf1/Lef1 TFs deployed mechanisms at multiple levels to ensure positive regulation of key targets in mature CD8$^+$ T cells to enforce T cell identity.

It is of note that a fraction of genes associated with WT-specific hubs showed elevated expression, such as *Gata3*, *Havcr2*, *Il12rb2*, *Islr*, *Ldlrad3,* and *Syk* (Fig. 7e, quadrant ii), suggesting that Tcf1/

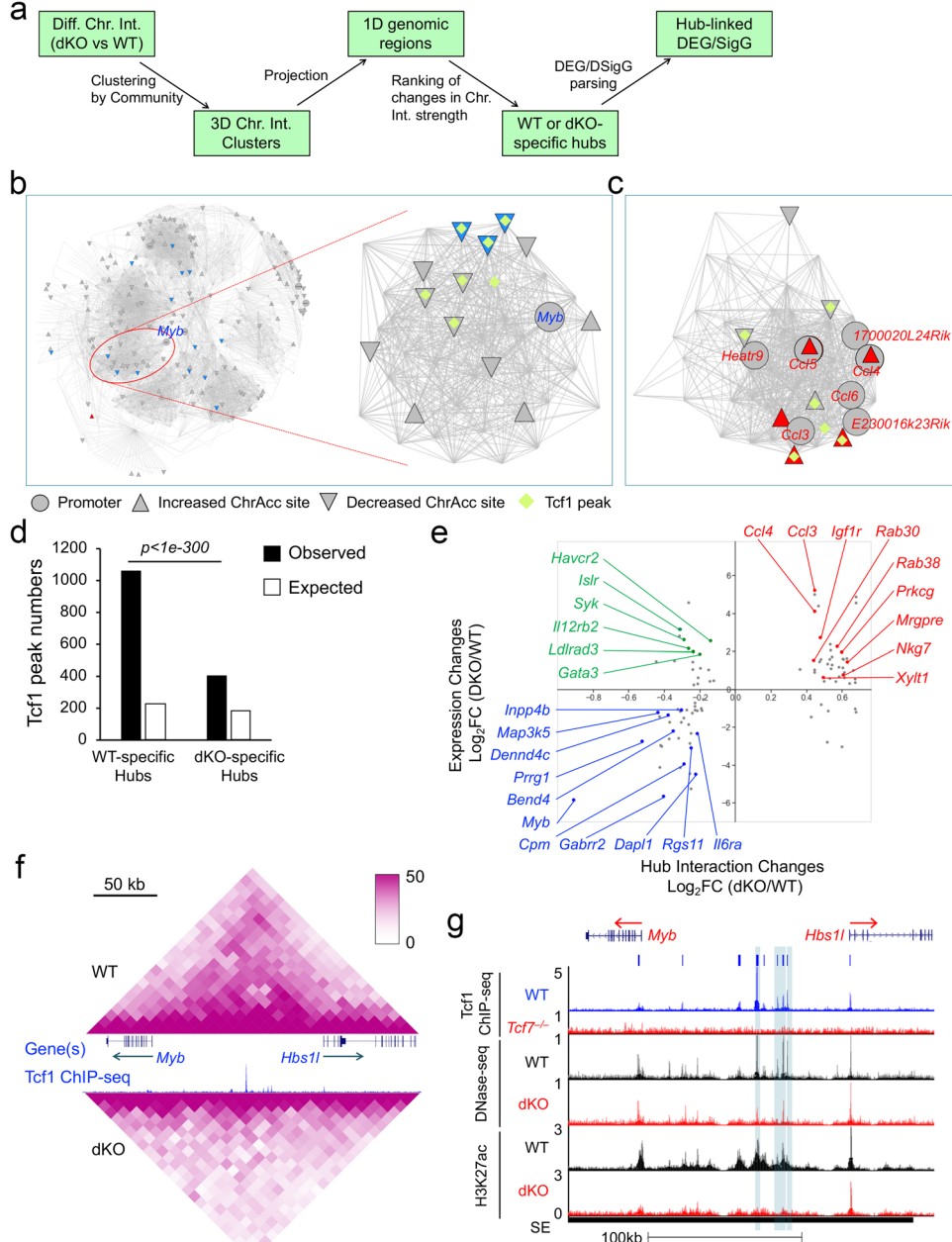

**Fig. 7 Tcf1/Lef1 TFs promote formation of chromatin interaction hubs for target gene transactivation. a** A diagram showing workflow of identifying WT- and dKO-specific chromatin interaction (Chr. Int.) hubs and hub-associated target genes. **b** Visualization of the output from chromatin interaction hub analysis. Shown on the left is the network containing decreased chromatin interactions connected to the *Myb* promoter (up to depth three nearest neighbors), where the nodes represent 10-kb bins on chromosome 10 and the lines represent chromatin interactions decreased in dKO CD8+ T cells. On the right is the WT-specific hub harboring the *Myb* gene. **c** A cluster on chromosome 11 harboring *Ccl* gene loci formed a dKO-specific hub. In (**b**) and (**c**), gene symbols are marked at the promoters (circle), with blue and red font denoting down- and upregulated expression in dKO CD8+ T cells, respectively. Triangles filled with blue and red denote statistically significant decrease and increase in ChrAcc in dKO CD8+ T cells, respectively. **d** Tcf1 peaks are more enriched in WT-specific hubs compared with dKO-specific hubs. The expected numbers of Tcf1 peaks were estimated based on genome average, and statistical significance is calculated using the one-sided Poisson test. **e** Scatterplot showing correlation between the changes of interaction strength in differential chromatin interaction hubs and expression changes of their associated genes, with selected genes marked. The x-axis value of a dot represents the median of all interaction changes in the corresponding hub. **f** Two-dimensional display of changes in chromatin interaction within the *Myb*-containing hub in WT and dKO CD8+ T cells. **g** One-dimensional display of changes in ChrAcc and H3K27ac within the *Myb*-containing hub in WT and dKO CD8+ T cells. Blue vertical bars denote ChrAcc sites showing reduction in dKO cells.

Lef1-mediated chromatin interactions could exert repressive function on transcription. In line with this notion, *Havcr2*, *Il12rb2*, *Ldlrad3*, and *Syk* genes were linked with Diff ChrAcc Cluster 2 (i.e., decreased ChrAcc sites coupled with increased gene expression in dKO CD8+ T cells, Fig. 5a; one example of

such sites can be found in the *Syk* intron, Fig. 5d), where the ChrAcc sites may function as transcriptional repressors. As a highlight of this point, the *Ldlrad3* (a pDC or monocyte LEG) gene locus was in a WT-specific hub formed on a chromosome 2 segment (Fig. 8a), and its gene body and downstream region

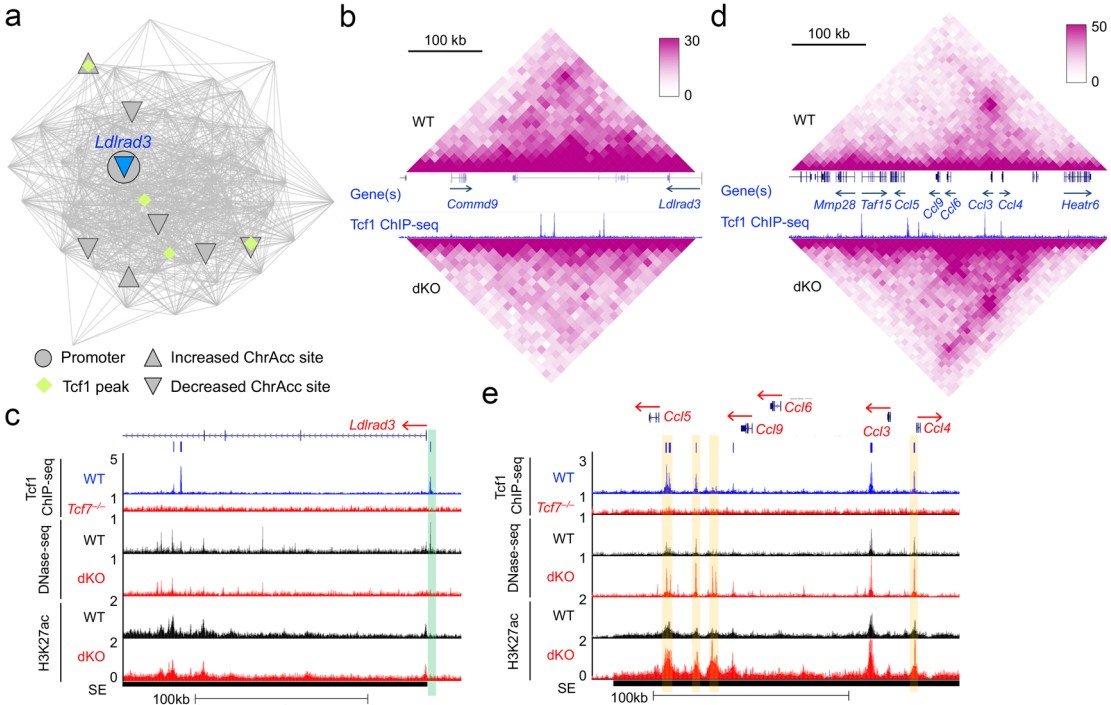

**Fig. 8 Tcf1/Lef1 TFs repress CD8+ lineage-inappropriate genes through modulating chromatin interactions. a–c** Tcf1 and Lef1 repress *Ldlrad3* by promoting chromatin interactions. **a** network display of a WT-specific hub containing the *Ldlrad3* gene. **b** 2D display of changes in chromatin interactions within the *Ldlrad3*-containing hub in WT and dKO CD8+ T cells. **c** 1D display of changes in ChrAcc and H3K27ac within the *Ldlrad3*-containing hub in WT and dKO CD8+ T cells. Green vertical bar denotes a ChrAcc site showing reduction in dKO cells. **d**, **e** Tcf1 and Lef1 repress *Ccl* genes by disengaging chromatin interactions, with the network display of a dKO-specific hub containing the *Ccl* genes in Fig. 7c. **d** 2D display of changes in chromatin interactions within the *Ccl* gene-containing hub in WT and dKO CD8+ T cells. **e** 1D display of changes in ChrAcc and H3K27ac within the *Ccl* gene-containing hub in WT and dKO CD8+ T cells. Yellow vertical bars denote ChrAcc sites showing increase in dKO cells.

showed decreased chromatin interactions in dKO CD8+ T cells (Fig. 8b). Additionally, the *Ldlrad3* hub contained a Cluster 2 ChrAcc site in the proximity of *Ldlrad3* promoter (Fig. 8a), which was found at 2.4 kb upstream of *Ldlrad3* TSS (Fig. 8c). These data underscore the integrative power of network analysis, suggesting that Tcf1/Lef1 TFs could promote chromatin accessibility and interactions to achieve repression of lineage-inappropriate genes in mature CD8+ T cells.

On the other hand, dKO-specific interaction hubs were more frequently associated with Tcf1/Lef1-repressed genes, which include many non-T cell LEGs such as *Igf1r* (granulocyte), *Xylt1* (monocyte), *Rab38* (cDC), *Prkcg*, and *Mrgpre* (pDC), *Rab30* (B cell), and *Nkg7* (NK cell LEG) (Fig. 7e, quadrant *i*). Of particular interest was a dKO-specific hub that contained multiple CCL genes including *Ccl3*, *Ccl4*, *Ccl6*, and *Ccl5* (defined as NK, γδT, and effector CD8+ T cell LEGs) (Fig. 7c). In this hub, a >200 kb region exhibited extensive increase in chromatin interactions in dKO CD8+ T cells, including regions with direct contact with the *Ccl* gene promoters (Fig. 8d). In addition, several dKO-specific ChrAcc sites were linked to the *Ccl* genes, including one at the *Ccl4* promoter and several upstream of *Ccl5* (Fig. 8e). The *Ccl* genes were all associated with a single SE with elevated H3K27ac in dKO CD8+ T cells (Fig. 8e). The *Ccl*-linked hub contained multiple Tcf1 peaks; nonetheless, none of these peaks contained Tcf/Lef consensus motifs. These observations suggest that Tcf1/Lef1 TFs is involved, but maybe through indirect mechanisms, in restraining chromatin accessibility and/or disengaging chromatin interactions to repress expression of CD8+ lineage-inappropriate genes.

**Tcf1 and Lef1 control multiple aspects of CD8+ T cell functionality.** Detailed molecular analyses above identified Tcf1/Lef1

target genes in mature CD8+ T cells and the underlying regulatory mechanisms utilized by the TFs. To determine how Tcf1/Lef1 control CD8+ T cell biology through their downstream transcription program, we validated protein expression changes in dKO CD8+ T cells for select target genes and then investigated their functional link with T cell biology. Tcf1/Lef1 deficiency diminished SE activity at the *Ccr7* gene locus (Fig. 6e), and dKO CD8+ T cells exhibited greatly reduced CCR7 expression (Fig. 9a, following gating strategy in Supplementary Fig. 8a). CCR7 is essential for migration of mature T cells to secondary lymphoid organs; indeed, when CD45.1+ WT and CD45.2+GFP+ dKO CD8+ T cells were adoptively transferred into a new host in 1:1 mixture, the dKO cells showed impaired capacity of homing to lymph nodes (Fig. 9b). dKO CD8+ T cells showed reduced Eomes transcripts (Supplementary Fig. 5b) as well as protein (Fig. 9c). Consistent with an important function of Eomes in generation and persistence of memory CD8+ T cells elicited by acute infection[49,50], the CD44hiCD122+ memory-phenotype CD8+ T cells were substantially diminished in uninfected dKO mice (Fig. 9d). dKO CD8+ T cells exhibited an enrichment of cytotoxic genes associated with effector CD8+ T cells (Supplementary Fig. 6c), resulting from a multitude of mechanistic actions by Tcf1/Lef1, including regulating a silencer at *Prdm1* locus (Fig. 6b, c), ChrAcc at *Gzmb* locus (Fig. 5b), SE and chromatin interaction at a cluster of *Ccl* gene loci (Fig. 8d, e). Indeed, the basal expression of granzyme B was elevated in dKO CD8+ T cells (Supplementary Fig. 8b), and CCL5 production showed pronounced increase in stimulated dKO CD8+ T cells (Supplementary Fig. 8c). After confirming that dKO CD8+ T cells were not more prone to apoptosis than WT cells after isolated ex vivo (Supplementary Fig. 8d), we stimulated the cells with titrating amounts of anti-CD3 and anti-CD28 using an in vitro culture system. Both WT and

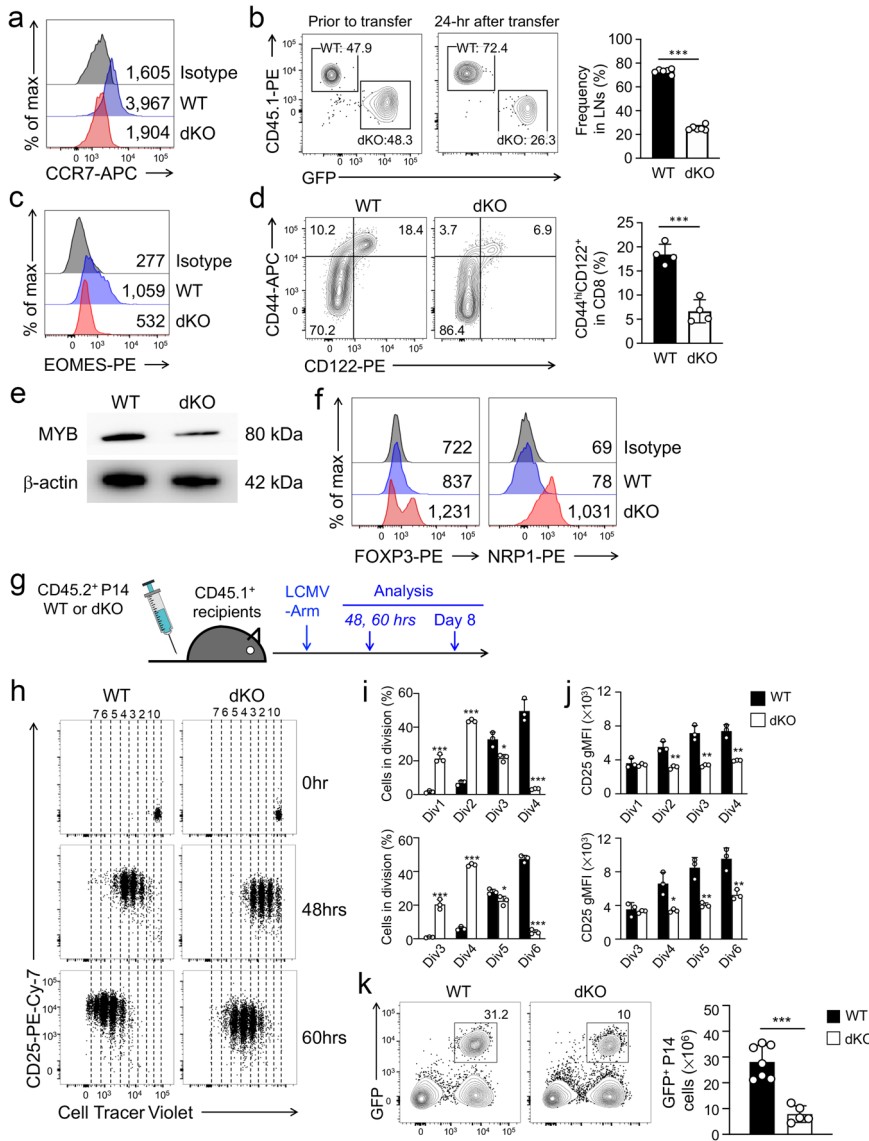

**Fig. 9 Tcf1/Lef1 TFs control multiple aspects of mature CD8+ T cell functions. a** Detection of CCR7 expression with cell-surface staining. Values denote geometric mean fluorescent intensity (gMFI). **b** T cell homing assay. CTV-labeled CD45.1+ WT and CD45.2+GFP+ dKO CD8+ T cells were mixed at 1:1 ratio (left) and transferred into CD45.2+ recipients, and 24 hrs later, donor cells were detected in recipient LNs (right). Cumulative data on donor cell abundance are means ± s.d. from two experiments. **c** Detection of EOMES expression with intranuclear staining, with gMFI marked. **d** Detection of memory-phenotype cells in splenic GFP+CD8+ T cells. Cumulative data on the frequency of CD44hiCD122+ cells are means ± s.d. from two experiments. **e, f** Detection of MYB expression with immunoblotting (**e**) and FOXP3 and NRP1 expression with flow cytometry (**f**). Data are representative from two experiments. **g** Experimental design for testing CD8+ T cell activation and differentiation in vivo. **h–j** Detection of CD8+ T cell division and CD25 induction during the first 60 hrs of activation. CTV-labeled splenic CD8+ T cells were transferred into CD45.1+ hosts followed by LCMV-Armstrong infection. CTV dilution and CD25 expression were monitored before (0 hrs), 48 or 60 hrs after infection in recipient spleens. **h** representative dot plots where dotted lines mark cells in different divisions. **i** Cumulative data on the frequency of cells in indicated cell divisions at 48 (top) and 60 hrs (bottom) post infection. **j**. Cumulative data on CD25 gMFI in cells at different dividing stages at 48 (top) and 60 hrs (bottom) post infection. Data are means ± s.d. from two experiments. **k** Detection of effector P14 CD8+ T cells on day 8 post infection. Effector P14 cells were detected as GFP+CD8+ T cells in recipient spleens, with values in representative contour plots denoting cell frequency. Cumulative data on effector P14 cell counts are means ± s.d. from two experiments. Statistical significance for (**b**, **d**, **i** and **k**) was assessed with two-tailed Student's t-test. *, $p < 0.05$; **, $p < 0.01$; ***, $p < 0.001$.

dKO CD8+ T cells showed similar capacity of producing IFN-γ and Granzyme B at all doses after 72-hr stimulation (Supplementary Fig. 8e). These observations suggest that the increased expression of effector genes in dKO CD8+ T cells was not translated into lower threshold of activation.

Owing to Tcf1/Lef1-mediated concerted regulation of chromatin interaction, SE activity and ChrAcc at the *Myb* locus

(Fig. 7), dKO CD8+ T cells showed potent downregulation of *Myb* transcripts (Supplementary Fig. 5b) and MYB protein (Fig. 8e). In addition, Treg lineage genes, including *Foxp3* and *Nrp1*, were enriched in dKO CD8+ T cells (Supplementary Fig. 6c), and both FOXP3 and NRP1 proteins showed elevated expression in dKO CD8+ T cells (Fig. 9f). MYB is necessary for effector CD8+ T cell expansion in response to viral infection[51],

and FOXP3 has broad transcription-suppressive functions that are even extrapolatable to nonlymphoid cells[52]. To further test how these transcriptomic and phenotypic changes due to Tcf1/Lef1 deficiency affected CD8$^+$ T cell-mediated immune response in vivo, we crossed P14 TCR transgene, which encodes a TCR specific for the LCMV GP33 epitope, onto the dKO background. The resulting P14 CD8$^+$ T cells were adoptively transferred into CD45.1$^+$ congenic recipients, followed by LCMV infection (Fig. 9g). By monitoring dilution of cell trace violet (CTV) during the initial 60 hrs post infection, we found that WT P14 cells exhibited rapid induction of CD25 and active cell division, with most of cells at 3–4 divisions by 48 h and at 5–6 divisions by 60 h (Fig. 9h–j). dKO P14 cells were activated, but showed slower rate of cell division at these early timepoints, with most of cells at the 2nd division by 48 h and at the 4th division by 60 hrs (Fig. 9h–j). The level of CD25 induction was also modestly reduced in dKO cells (Fig. 9h, j). Consistent with the changes at these early timepoints, on day 8 post infection when CD8$^+$ T cells reached peak response, the expansion of dKO effector P14 cells was reduced by 2/3 compared with WT cells (Fig. 9k). In addition, dKO effector cells showed moderately diminished polyfunctionality in producing IFN-γ together with TNF and IL-2 (Supplementary Fig. 8f, 8g). Collectively, these data demonstrated that ablating Tcf1/Lef1 in naïve CD8$^+$ T cells compromised their functions in multiple aspects at the homeostatic state and in response to pathogen challenge.

## Discussion

T cell identity is established during thymic development, under the direction of key TFs including Tcf1. Maintenance of T cell identity, in at least some aspects, such as *Cd4* gene silencing, is thought to be autonomous through epigenetic mechanisms[19]. In this study, specific ablation of Tcf1 and Lef1 in mature CD8$^+$ T cells resulted in not only diminished expression of T cell lineage-enriched genes, but also aberrantly upregulated genes that were enriched in non-T cell lineages including B cells, DCs and monocytes. In addition, Tcf1/Lef1 TFs remain necessary to maintain CD8$^+$ T cells at their naïve state before encountering cognate antigens by repressing cytotoxic genes induced in effector CD8$^+$ T cells. These data demonstrate that CD8$^+$ T cells require constant supervision of their unique identity and naïve state, and Tcf1/Lef1 TFs have essential functions in these critical processes.

In line with a general feature of HMG family proteins, Tcf1/Lef1 TFs have been shown to bend DNA in vitro and postulated to have direct impact on DNA structures in the 3D space[22,23]. Comprehensive analysis of Hi-C profiles of WT and Tcf1/Lef1-deficient CD8$^+$ T cells showed that Tcf1/Lef1 TFs modulates genomic organization on multiple scales including A/B compartments, TADs, hubs and focal chromatin looping, with stronger impact observed at genomic regions with higher density of Tcf1 binding peaks, especially those with direct Tcf1 binding. The direct impact on focal chromatin loops by Tcf1/Lef1 TFs is in line with the punctate nature of Tcf1 binding peaks on the genome. In addition to enriched Tcf1 occupancy at anchors of chromatin loops, extensive Tcf1 binding peaks were found within TADs. Our integrative network analysis revealed quantifiable impact of Tcf1/Lef1 TFs on chromatin interactions, demonstrating wide-spread reduction in contact frequency in Tcf1/Lef1-deficient CD8$^+$ T cells within chromatin interaction hubs. Thus, Tcf1/Lef1 TFs help form highly connected contact matrices to facilitate interactions among neighboring regulatory regions, consistent with their DNA-bending capacity. This observation is compatible with a trending concept that weak but multivalent interactions among macromolecules, such as TFs and their

cofactors, form phase-separated condensates for highly coordinated transcriptional control[53,54].

Complementing the larger scale of Hi-C data, systemic mapping of chromatin accessibility and super enhancer activity showed their coordinated actions with chromatin looping in CD8$^+$ T cells. Tcf1 is considered a 'pioneer' factor that establishes ChrAcc landscape during T cell development[29–31]. In mature CD8$^+$ T cells, loss of Tcf1 and Lef1 resulted in substantial changes in ChrAcc. Through detailed motif analysis of Tcf1 binding peaks, we found that Tcf1 direct binding sites, where Tcf1 has direct contact with DNA elements, overlapped extensively with ChrAcc sites showing reduced accessibility in Tcf1/Lef1-deficient CD8$^+$ T cells. This finding indicates that the immediate, direct impact by Tcf1/Lef1 TFs on the chromatin is to maintain the ChrAcc at an open state in CD8$^+$ T cells. Importantly, changes in ChrAcc state and super enhancer activity are concordant with the strength of chromatin loops and interactions, suggesting that Tcf1/Lef1 TFs function as key orchestrators of chromatin open state at cis-element levels, super enhancer activity on a larger scale, and genomic organization in 3D space.

As frequently demonstrated, global changes in chromatin state and genomic organization are not always directly consequential in altering transcriptional output. We took a gene-centric approach, i.e., focusing on DEGs and differentially expressed immune cell lineage-enriched genes, which proved to be mechanistically informative. At Motif$^+$ Tcf1 direct binding sites, Tcf1/Lef1 TFs maintain chromatin at open state and promote chromatin interactions. It is not surprising that these regulatory effects resulted in positive regulation of their downstream target genes including key TFs such as *Myb*. However, the ChrAcc and/or chromatin interactions supported by Tcf1/Lef1 TFs also showed repressive effects depending on the gene loci, as exemplified at the Tcf1-bound, –24 kb element upstream of the *Prdm1* gene. Notably, this upstream *Prdm1* element lacked H3K27ac modification, in key contrast to an active enhancer in *Prdm1* intron 3. These observations suggest the versatility of Tcf1/Lef1 TFs in implementing gene regulation. They directly determine chromatin state and/or interactions, and cooperate with cofactors such as histone modifiers/readers in specific gene context, to achieve an output of either transcriptional activation or repression.

As evident in our systematic molecular analyses, Tcf1 binding events are associated with distinct regulatory effects, such as promoting or disengaging chromatin interaction, increasing or reducing chromatin accessibility, and activating or repressing target gene expression, depending on the gene context. By use of position-weight matrix in motif analysis, we made the distinction between Tcf1 direct vs. indirect binding events and assessed their relative contribution to each regulatory mechanism. At its direct binding sites, Tcf1 exhibited clearly distinguishable preference for promoting chromatin interactions and maintaining chromatin at an open status. Such distinction predicts likelihood of a preferred functional outcome, but should not be interpreted in absolute terms. Besides the direct, positive impact by Tcf1/Lef1 TFs on chromatin, we noted that ablation of Tcf1 and Lef1 in mature CD8$^+$ T cells resulted in increased ChrAcc, elevated super enhancer activity, and aberrantly enhanced chromatin interactions at distinct gene loci and/or genomic regions. These changes were associated with abnormal induction of lineage-inappropriate, or effector-associated genes. Whereas scores of Tcf1 peaks could be found in super enhancers, chromatin loop anchors and interaction hubs because of their large sizes, the relative enrichment of Tcf1 binding was substantially lower in these large structures specific to Tcf1/Lef1-deficient CD8$^+$ T cells than in those specific to WT cells. On a finer scale, ChrAcc sites specific to Tcf1/Lef1-deficient CD8$^+$ T cells rarely overlapped with strong

Tcf1 binding peaks, and the overlapping Tcf1 peaks rarely contained Tcf/Lef motifs. Therefore, Tcf1/Lef1 TFs could be 'directly' involved, but through 'indirect' mechanisms, in the repressive functions such as constraining ChrAcc and disengaging chromatin interactions. For example, due to the lack of Tcf/Lef motifs at those Tcf1 peaks, Tcf1 is likely 'indirectly' recruited by other TFs that have direct contact with DNA, to serve as a key component in protein complexes exerting repressive actions. In this regard, we previously demonstrated that Tcf1 and Lef1 have intrinsic histone deacetylase activity[17], and in immunization-elicited follicular helper T cells, Tcf1 binds to the *Ctla4* gene locus indirectly to restrain its ChrAcc[55]. Other possible mechanisms may involve aberrantly induced proteins and/or increased availability of Tcf1/Lef1 cofactors in Tcf1/Lef1-deficient cells, which in turn promote ChrAcc or chromatin interactions. These possibilities warrant detailed molecular dissection in future investigations.

It should be noted that resolution remains a major limiting factor for Hi-C-based analysis of enhancer–promoter linkage, especially for short-range interactions[12,13]. In this study, we reached 10-kb resolution, which remained lower than what was achieved for DNase-seq, H3K27ac and Tcf1 ChIP-seq. As a result, many known Tcf1/Lef1 downstream genes did not register as being regulated through the structural mechanism. For example, the *Prdm1* gene encompasses a 20.4 kb genomic region, and the upstream silencer, identified in this work based on Tcf1/Lef1-dependent ChrAcc changes, was 24 kb from the *Prdm1* TSS. Resolution at a finer scale is needed to definitively resolve the short-range interactions among *Prdm1* promoter, its intronic enhancer and upstream silencer. Nonetheless, our analysis of super enhancers and genomic architecture on a larger scale did identify Tcf1/Lef1 target genes with profound implications in CD8+ T cell biology. These included Tcf1/Lef1-mediated positive regulation of *Ccr7* and *Myb*, which underlie the homing and proliferative capacity of CD8+ T cells, respectively. In fact, the diminished proliferation and poly-cytokine production capability in Tcf1/Lef1-deficient effector CD8+ T cells were a phenocopy of defects observed in Myb-ablated CD8+ T cells[51].

In summary, this study demonstrates that Tcf1/Lef1 TFs directly regulate global genomic organization on multiple scales in mature CD8+ T cells. By adapting network analysis, our systematic approaches enhance integration of Hi-C and multiomics data and show concordant modulation of chromatin accessibility, super enhancer activity, chromatin looping and chromatin interaction hubs by Tcf1/Lef1 TFs. These approaches help establish direct linkage of these multifaceted mechanisms to transcriptional output at immune cell lineage-enriched genes, showing an important function of Tcf1/Lef1 TFs in supervising mature CD8+ T cell identity to prevent lineage confusion and protecting critical CD8+ T cell functions. Our findings also provide important insights into the interplay between TFs and 3D genome architecture in general, highlighting the integrative nature of TFs' actions on transcriptional regulation.

## Methods

**Mice**. Rosa26GFP knock-in mice, *Cd4-Cre,* and *hCD2-Cre* transgenic mice were from the Jackson Laboratory. *Tcf7*fl/fl and *Lef1*fl/fl mice were previously reported[56,57]. All strains were generated on the C57BL/6 background or backcrossed to this background for ≥10 generations. All compound mouse strains used in this work were from in-house breeding at the animal care facilities of University of Iowa and Center for Discovery and Innovation, Hackensack University Medical Center. All mice analyzed were 6–12 weeks of age, and both genders were used without randomization or blinding. All the procedures using mice were in compliance with all relevant ethical regulations for animal testing and research. All mouse experiments received ethical approval from the Institutional Animal Use and Care Committees of the University of Iowa and Center for Discovery and Innovation, Hackensack University Medical Center.

**Cell isolation and flow cytometry**. Because excision efficiency by *hCD2-Cre* ranged from 80-95%, *Rosa26*GFP allele was used to mark target gene-ablated cells. All analyses in this work were performed on GFP+TCRβ+CD8+ T cells in CD62L+CD44lo-med naïve state from *hCD2-Cre*+ *Rosa26*GFP*Tcf7*+/+*Lef1*+/+ (WT) and *hCD2-Cre*+*Rosa26*GFP *Tcf7*fl/fl*Lef1*fl/fl (dKO) mice. The fluorochrome-conjugated antibodies, including anti-CD8 (53-6.7), anti-TCRβ (H57-597), anti-CD4 (RM4-5), anti-CD44 (IM7), and anti-CD62L (MEL-14), anti-CD45.1 (A20), anti-CD45.2 (104), anti-Granzyme B (GB12), anti-IFN-γ (XMG1.2), anti-TNF (MP6-XT22), anti-EOMES (Dan11mag), anti-CD122 (TM-β1), anti-CD25 (PC61.5), anti-NRP1(3DS304M) were from eBiosciences, Thermo Fisher Scientific, and anti-CCR7 (4B12) was from BioLegend.

Single-cell suspensions from the spleen and lymph nodes (LNs) were generated after mashing tissue through 70 µm cell strainer. For the detection of cytokines, the cells were stimulated with GP33 peptides (200 nM) or PMA (50 ng/ml) and ionomycin (500 ng/ml) in the presence of Golgi Stop and Golgi Plug for 5 h at 37 ℃, followed by standard intracellular staining. For intranuclear detection of EOMES, Tcf1 or FOXP3, surface-stained cells were fixed and permeabilized with the Foxp3/Transcription Factor Staining Buffer Set (eBiosciences, Thermo Fisher Scientific), followed by incubation with corresponding fluorochrome-conjugated antibodies. All antibodies are diluted at 1:200 and the cells were stained at 4 ℃ for 30 min. For detection of cell survival status, the PE Annexin V Apoptosis Detection Kit (BD Biosciences, Cat. No. 559763) was used following the manufacturer's instruction. Cell sorting was performed on FACSAria (BD Biosciences). Data were collected on Fortessa LSR flow cytometer (BD Biosciences) and were analyzed with FlowJo software v10.7.1 (TreeStar).

**High-resolution chromosome-conformation-capture (Hi-C)**

*Hi-C*. Hi-C was performed using the three enzyme Hi-C (3e Hi-C) approach as described[58]. WT and dKO naïve CD8+ T cells (each in two replicates, $4 \times 10^6$ cells/replicate) were sorted and crosslinked with 1% formaldehyde for 10 min at 25 ℃. The crosslinked cells were lysed in 10 ml lysis buffer (10 mM Tris-HCl pH 8.0, 10 mM NaCl, 0.2% NP-40) supplemented with protease inhibitor cocktail (MilliporeSigma) at 4 ℃ for 1 hr. The nuclei were collected and treated with 400 µl 1× CutSmart buffer (NEB) containing 0.1% SDS at 65 ℃ for 10 min, and TritonX-100 was added to a final concentration of 1% to quench SDS. The resulting chromatin was then digested with three restriction enzymes, CviQ I, CviA II, and Bfa I (NEB), at 20 units each at 37 ℃ for 20 min. The reaction was stopped by washing with 600 µl wash buffer (10 mM NaCl, 1 mM EDTA, 0.1% Triton X-100) two times. The DNA ends were blunted and labeled with biotin by Klenow enzyme in the presence of dCTP, dGTP, dTTP, biotin-14-dATP, followed by ligation using T4 DNA ligase. After reverse crosslinking, DNA was fragmented by sonication with a Covaris S2 ultrasonicator. The DNA fragments were then end-repaired, and the biotinylated DNA fragments were captured using Dynabeads MyOne Streptavidin C1 beads (Invitrogen, Thermo Fisher Scientific). The DNA on beads was ligated to the Illumina Paired End Adapters, and amplified with PCR for library construction. DNA fragments of 300-700 bp were purified from 2% E-gel and sequenced on HiSeq4000 in paired read mode with the read length of 150 nucleotides. The Hi-C data are deposited at the GEO (GSE164710) under the SuperSeries of GSE164713.

*Hi-C library mapping*. Iterative_mapping from 25 bps to 105 bps with a step size of 5 bps using hiclib (https://github.com/mirnylab/hiclib-legacy) was applied to the Hi-C sequencing libraries for alignment onto reference genome mm9. Picard (http://broadinstitute.github.io/picard/) was then applied for redundancy removal. The resulting libraries were subject to further processing with Mirnylib with default parameters except filterDuplicates (mode = 'ram') (https://github.com/mirnylab/hiclib-legacy) into hdf5 file. The hdf5 files were converted into text files and then .hic files using the Juicer[30] pre function. The .hic file is a highly compressed binary file that provides rapid random access to the binned matrices at nine resolutions: 2.5m, 1m, 500k, 250k, 100k, 50k, 25k, 10k, and 5k base pairs.

*Measurement of reproducibility of Hi-C replicates*. The binned contact matrices were converted into a text file using the dump function in Juicer v1.21.01[30] with parameters (observed; delimited: base-pair; resolution:10 kb; normalization: KR). For each anchor and each replicate, the respective row sum of the contact matrix elements (excluding the diagonal element) was calculated. Scatterplots of the resulting data were used to calculate the Pearson correlation of the replicates.

*Detection and quantification of A/B compartments*. A/B compartments were identified using the Eigenvector function from Juicer v1.21.01[30] with parameters (delimited: base-pair; resolution: 100 kb; normalization: None). Then Eigenvector applies principal component analysis to extract the first principal component of the Pearson correlation matrix as the compartment scores (up to a sign). For each chromosome, the sign of the compartment score was fixed by comparing with the H3K27ac profile, with the expectation that A compartments are enriched with H3K27ac.

*Definition and quantification of topological associated domains*. Topological associated domains (TADs) were identified by the Arrowhead algorithm from Juicer v1.21.01[30] using the medium resolution maps (i.e., m: 2000; resolution: 10 kb;

normalization: KR). A total of 1,723 TADs were identified in WT naive CD8[+] T cells using the pooled Hi-C data. The TAD score is defined as the ratio of the total number of intra-TAD read pairs divided by the total number of read pairs that overlap with the TAD. The read pairs within the same 10-kb bin were excluded from counting.

*Calculation of interaction score of an anchor.* The binned contact matrices were converted into a text file using the dump function in Juicer v1.21.01[30] with parameters (observed; delimited: base-pair; resolution:10 kb; normalization: KR). For an anchor, the respective row sum of the contact matrix elements (excluding the diagonal element) was defined as the interaction score.

*Identification of chromatin loops.* Chromatin loops were identified using the HiCCUPS algorithm from Juicer v1.21.01[30] (CPU version; medium resolution maps) with significance threshold FDR = 0.1 and donut FDR = 1.0e−4. A total of 11,490 and 12,151 loops were identified from the pooled Hi-C libraries of WT and dKO CD8[+] T cells, respectively.

*Tcf1 enrichment in annotated chromatin loops.* Chromatin loops in WT CD8[+] T cells were allocated into three categories according to annotations of loop anchors. A loop is annotated as a promoter–promoter (PP) loop if both anchors overlap with promoters. A loop is annotated as a promoter-enhancer (PE) loop if one anchor overlaps with a promoter and the other anchor overlaps with a chromatin-accessible (ChrAcc) site (as measured with DNase-seq) but not any promoters. A loop is annotated as an enhancer–enhancer (EE) loop if both anchors overlap with ChrAcc sites but not promoters. A promoter is defined as the region around the transcription start site (TSS), [TSS -1kb, TSS +1kb]. Enrichment of Tcf1 binding was then calculated on the anchors of each category of loops (see Enrichment analysis of *Tcf1* binding).

*Identification of hubs of chromatin loops.* The chromatin loops in WT CD8[+] T cells were analyzed from a network perspective using the igraph platform[33], where the nodes of the network correspond to anchors of the loops, and the network edges correspond to loops (Supplementary Fig. 3d). The clusters within the network were identified using the community_multilevel algorithm[59] with parameters "weights" = 1 for each loop and "return_levels" = 1. Clusters were ranked according to size, and hubs were identified as top-ranking clusters above a threshold. To set a proper threshold, we adopted an approach motivated by the identification of super enhancers[39]. Briefly, we ranked the clusters by their sizes and plotted the size of each cluster against its ranking. The data were then scaled so that both x- and y-axis values range from 0 to 1. The threshold was identified as the x-axis point for which a line with a slope of one was tangent to the curve. Clusters whose size above the threshold were identified as hubs. For loops in the WT CD8[+] T cells, the threshold was found to be four. In each hub, the anchors were then ranked according to the interaction frequency using the PageRank algorithm. The top anchor and the bottom anchor for each hub were collected for comparison of Tcf1 binding enrichment.

*Identification of differential chromatin loops.* Chromatin loops were called in WT and DKO CD8[+] T cells separately using the pooled Hi-C data as described above and then combined as a union chromatin loop pool. The number of read pairs associated with the anchor pairs of the union loops was counted to generate a count matrix. The count matrix, which includes two biological replicates from both WT and dKO CD8[+] T cells, was used as input for edgeR (v.3.20.7.2)[60] (quasi-likelihood test, robust) with requirements of fold-change >2.0 and p value <0.05 to identify significant differential loops between WT and dKO CD8[+] T cells. A total of 877 and 1,682 WT- and dKO-specific chromatin loops were identified, respectively.

*Hubs of differential chromatin interactions and their linked genes.* To identify hubs of differential chromatin interactions, the contact matrices (resolution:10 kb; normalization: KR) of WT and dKO CD8[+] T cells were compared (Fig. 7a). To identify WT-specific hubs, chromatin interactions that showed decrease in the dKO CD8[+] T cells were selected to construct a network using the igraph platform[33]. Interactions within the same bin and weak interactions. i.e., (WT + DKO) ≤ 10, were excluded. Clusters on the network were identified using the community_multilevel algorithm[59] (weights = WT interaction frequency, return_levels = 1). The clusters were then projected onto genomic sequence as follows: (1) in each cluster, its nodes were ranked by pagerank score, and the genomic bin of the node with the highest pagerank score was chosen as the seed of the projection; (2) for each of the rest of the nodes, its genomic bin was included in the projection only if (its degree)/(the distance (in units of bins) between its genomic location to the seed)^2 < 0.5. After the projection, each cluster in the network was projected onto a continuous genomic region. For each of the projected regions, one-sided Wilcoxon signed rank test was used to calculate the significance of differences of internal interactions between WT and dKO CD8[+] T cells. Only regions with p value <1.0e−7 were considered as WT-specific domains. The dKO-specific hubs were identified following the same procedure by focusing on chromatin interactions that showed increase in the dKO CD8[+] T cells. A gene was associated with a WT- or dKO-specific hub if its promoter was within the hub.

## Tcf1 ChIP-seq and data analyses

*Tcf1 ChIP-seq.* Recombinant full-length Tcf1 protein was purified and used to immunize rabbits to obtain Tcf1 antisera, so as to increase the probability of capturing exposed Tcf1 epitopes especially in large protein complexes. Naïve CD8[+] T cells were sort-purified from WT or Cd4-Cre[+]Tcf7[fl/fl] (Tcf7[−/−]) mice, fixed and lysed[17]. The nuclei were sonicated with a Q125 sonicator equipped with an 1/8-inch diameter probe (QSonica) at 20% input amplitude, at a 20-second duration for eight times. The resulting chromatin fragments were immunoprecipitated with the anti-Tcf1 antiserum, properly washed, and library constructed following standard protocols[17]. The libraries were sequenced on HiSeq2000 in paired read mode with a read length of 150 nucleotides. The Tcf1 ChIP-seq data are deposited at the GEO (GSE164670) under the SuperSeries of GSE164713.

*Tcf1 ChIP-seq data processing.* The sequencing quality of ChIP-Seq libraries was assessed by FastQC v0.11.4 (http://www.bioinformatics.babraham.ac.uk/projects/fastqc/). Bowtie2 v2.2.5[61] was used to align the sequencing reads to the mm9 mouse genome. For Tcf1 ChIP-seq in WT naive CD8[+] T cells, MACS v2.1.1[26] was used for peak calling with Tcf1 ChIP-seq in Tcf7[−/−] CD8[+] T cells as a negative control, under a stringent statistical criteria of fold enrichment ≥4, p value <1e−5 and FDR < 0.05. A total of 19,042 high-confidence Tcf1 binding peaks were identified.

*Tcf1 motif analysis.* For motif analysis of the identified Tcf1 peaks, the sequences of ±100 bps flanking the peak summits were used as input to findMotifsGenome.pl from HOMER for de novo motif discover. For motif scanning, we used the core sequence (with consensus sequence ATCAAAG) of the MA0769.1 from HOMER, which is represented by a position weight matrix. The sequence underlying each Tcf1 peak is scored by computing the log (odds ratio). A Tcf1 peak was considered Motif[+] if it contained a motif hit with log (odds ratio) >7 and Motif[−] if all of its motif hits had log (odds ratio) <3.

*Enrichment analysis of Tcf1 binding.* On a set of regions compared to genome background, the enrichment score of Tcf1 was calculated as the observed number of binding events divided by the expected number of binding events. The expected number of Tcf1 binding events was calculated by multiplying the total length of the regions with the genome-wide density of binding events. The statistical significance of the enrichment was calculated using a one-sided binomial test provided in the Scipy package (v1.5.2), where the total number of Tcf1 binding events was used for "number of trials", the number of observed Tcf1 binding was used for the "number of successes" and the ratio of expected number over the total number of events was used for the "probability".

## H3K27ac ChIP-seq and data analyses

*H3K27ac ChIP-seq.* WT or dKO naïve CD8[+] T cells were sorted in 4 and 3 replicates, respectively. The cells were fixed, and chromatin segments were immunoprecipitated with anti-H3K27ac (Abcam, ab4729) followed by Hi-Seq sequencing following standard procesures[17]. The H3K27ac ChIP-seq data are deposited at the GEO (GSE164711) under the SuperSeries of GSE164713.

*H3K27ac ChIP-seq data processing.* The sequencing quality of ChIP-seq libraries was assessed by FastQC v0.11.4 (http://www.bioinformatics.babraham.ac.uk/projects/fastqc/), and sequencing reads were aligned to the mm9 mouse genome as described above, and only uniquely mapped reads (MAPQ > 10) were retained. Mapped reads from replicates were pooled for WT or DKO CD8[+] T cells, and were processed with SICER (v1.1)[40] for H3K27ac peak calling with the setting of FDR < 0.01, windows size = 200 bps, and gap size = 400 bps. A total of 65,563 and 82,213 H3K27ac peaks were identified in the pooled H3K27ac libraries for WT and dKO CD8[+] T cells, respectively, and these H3K27ac peaks were combined to generate a union H3K27ac peak pool containing 76,633 unique H3K27ac peaks. In reproducibility analysis, the reads on each of the 76,633 union peaks were counted in each library, and then normalized by the total read counts on the union peaks in each library. The resulting matrix was used for the principal component analysis.

*Identification of super enhancers.* Super enhancers (SEs) were identified by applying the Ranking Ordering of Super Enhancer (ROSE) algorithm[39], which stitches neighboring H3K27ac islands called by SICER in continuous genomic regions. A total of 1,160 and 980 SEs were identified in WT and dKO CD8[+] T cells, respectively.

*Identifications of differential super enhancers.* SEs identified in WT and dKO CD8[+] T cells were merged to generate a union SE pool that contained a total of 1,190 SE regions. The reads from individual replicates of WT and dKO CD8[+] T cells were counted on these regions, and the resulting read count matrix was used as input for edgeR (v.3.20.7.2)[60] (quasi-likelihood test, robust) with a statistical stringency of FDR < 0.05 to identify WT- and dKO-prepotent SEs. Because the "stitching" brought in H3K27ac reads that were not on H3K27ac islands into an SE, we extracted H3K27ac island-filtered reads and total reads to calculate a signal-to-noise ratio for each SE. The WT- and dKO-prepotent SEs had signal-to-noise ratio

at 0.9319 and 0.9327, respectively, hence confirming that the differential SE activity was not due to changes in background H3K27ac reads.

## DNase-seq and data analyses

*DNase-seq.* DNase-seq was performed following standard procesures[62]. In brief, WT or dKO naïve CD8[+] T cells were sorted in 3 and 2 replicates ($3 \times 10^5$ cells/replicate), respectively. The cells were lysed in lysis buffer (10 mM Tris-HCl pH 7.5, 10 mM NaCl, and 3 mM $MgCl_2$) and digested with 2.4 units of DNase I at 37 °C for 5 min. The reaction was terminated by addition of stop buffer (10 mM Tris-HCl pH7.5, 10 mM NaCl, 10 mM EDTA, 2% SDS, 0.5 mg/ml Proteinase K, and 1 ng/ml of circular carrier DNA), and incubated at 65 °C for 1 hr. After purification with phenol-chloroform extraction and ethanol precipitation, the DNA was end-repaired using End-It DNA-Repair kit (Lucigen, Cat. No. ER81050) at 37°C for 20 min, and then treated with Klenow fragment (3'–>5' exo–, NEB) and dATP to yield a protruding A base at the 3' end. The DNA fragments were then ligated to the Illumina Paired End Adapters, and amplified with PCR for library construction. PCR products between 160-300 bp were isolated on 2% E-gel for sequencing on Illumina HiSeq2000 in paired-end mode with the read length of 150 nucleotides. The DNase-seq data are deposited at the GEO (GSE164689) under the SuperSeries of GSE164713.

*DNase-seq data processing.* The sequencing quality of DNase-seq libraries was assessed by FastQC v0.11.4 (http://www.bioinformatics.babraham.ac.uk/projects/fastqc/). Bowtie2 v2.2.5[61] was used to align the sequencing reads to the mm9 mouse genome, and only uniquely mapped reads (MAPQ > 10) were retained. The mapped reads from multiple replicates were pooled for WT or DKO CD8[+] T cells, and were processed with MACS v2.1.1[26] for calling DNase I-hypersensitive peaks, with stringent criteria of ≥4 fold enrichment, $p$ value<1e−5 and FDR < 0.05. For consistency, the DNase I-hypersensitive peaks are referred to as chromatin-accessible (ChrAcc) sites in this work.

A total of 28,472 and 24,363 ChrAcc sites were identified in WT and dKO CD8[+] T cells, respectively, and these sites were merged to generate a union ChrAcc site pool that contained 28,827 unique ChrAcc sites. For reproducibility analysis, reads at each union site were counted in each DNase-seq library, and then normalized by the total read count of the union sites in respective library. The resulting matrix was used for principal component analysis. The pooled data were used for plotting the aggregated ChrAcc profiles at the Tcf1 peaks (Fig. 3c) with normalization by the total read counts of the union sites in each cell type.

*Identification of differential ChrAcc sites.* The reads on the 28,827 union ChrAcc sites were counted in each DNase-seq library of the WT and dKO replicates. The read count matrix was used as input for edgeR (v.3.20.7.2)[60] (quasi-likelihood test, robust, fold-change 2.0 and FDR < 0.05) to identify differential ChrAcc sites between WT and dKO CD8[+] T cells. A total of 987 WT-specific and 576 dKO-specific ChrAcc sites were identified, respectively.

*Association of regulatory elements to genes.* A regulatory element was associated with a gene if it was located within the regions from −50 kb upstream of transcription start site to the transcription end site. The remaining elements were associated with its closest gene using the "closest" function from Bedtools[63] (parameter: t'all'). Associations were also made between regulatory elements and gene promoters connected by chromatin loops called from the Hi-C data.

**RNA-seq and data analysis.** Total RNA was extracted from the sorted naïve WT or dKO CD8[+] T cells. cDNA synthesis and amplification were performed using SMARTer Ultra Low Input RNA Kit (Takara Bio, Cat. No. 634848), following standard procedures[64]. Libraries were sequenced on Illumina's HiSeq2000 in single-end mode with a read length of 50 nucleotides. The RNA-seq data are deposited at the GEO (GSE164712) under the SuperSeries of GSE164713.

The sequencing quality of RNA-seq libraries was assessed by FastQC (v0.11.4), and adapters were removed through Cutadapt[65]. The reads were mapped to mouse genome mm9 using Tophat (v2.1.0)[66]. Mapped reads were then processed by Cuffdiff (v2.2.1)[67] to estimate expression levels of all genes and identify differentially expressed genes (DEGs). The expression level of a gene was expressed as a gene-level Fragments Per Kilobase of transcripts per Million mapped reads (FPKM) value. The reproducibility of RNA-seq data was evaluated by applying the Principal Component Analysis for all genes between biological replicates. Upregulated or downregulated DEGs in CD8[+] T cells were identified by requiring ≥ 2-fold expression changes and FDR < 0.05, as well as FPKM ≥ 1 in higher expression samples. UCSC genes from the iGenome mouse mm9 assembly (http://support.illumina.com/sequencing/sequencing_software/igenome.html) were used for gene annotation.

**Identification of immune cell lineage-enriched genes and GSEA.** The transcriptomes of selected reference immune cells, i.e., the Normalized Gene Table, were downloaded from the Immunological Genome Project (http://www.immgen.org/) data browsers under entry GSE109125[42]. To define lineage-enriched genes (LEGs) for immune cells under homeostatic state, data from the following cell types were extracted: (1) pan-T cells including naïve CD4[+], naïve

CD8[+], Treg and γδT cells; (2) B cells (where the B.Sp#4 replicate was excluded because of its strong divergence from other replicates); (3) NK cells including CD27[+]CD11b[+], CD27[+]CD11b[−], and CD27[−]CD11b[+] subsets; (4) conventional DCs (cDCs) including CD4[+] and CD8[+] DC subsets; (5) plasmacytoid DCs (pDCs); (6) monocytes including Ly6C[+] and Ly6C[−] subsets; and (7) granulocytes. DEGs were first identified between each pair of the seven cell types using edgeR 3.28.1 (≥5-fold-change and FDR ≤ 0.01). Using the DEG matrix, LEGs for a cell type were identified as genes that were significantly upregulated in at least five out of six comparisons with the other cell types, with an additional requirement of standard deviation/means <1 across replicates within a specific cell type to remove outliers.

Within the T lineage cells at the homeostatic state, DEGs were identified among naïve CD4[+], naïve CD8[+], Treg and γδT cells through pairwise comparison as above. LEGs for a T cell subtype were required to be significantly upregulated in at least two out of three comparisons with the other T cell subtypes with an additional requirement of standard deviation/means <1 across replicates within a specific subtype to remove outliers.

These T cell subtypes were combined to compare with effector CD8[+] T cells to define the effector LEGs following the same criteria. The immune cell LEGs were collected as custom gene sets and used in GSEA[68] to measure their relative enrichment in WT and dKO CD8[+] T cell transcriptomes.

**Adoptive transfer and viral infection.** For adoptive transfer, naïve P14 CD8[+] T cells were isolated from the LNs from WT and dKO P14 TCR-transgenic mice. For evaluation of cell division at 48–60 hrs after activation, P14 CD8[+] T cells were labeled with 10 µM Cell Trace Violet (CTV, ThermoFisher Scientific, Cat. No. C34557), and $1 \times 10^6$ of CTV-labeled Vα2[+] P14 CD8[+] cells were adoptively transferred into CD45.1[+] B6.SJL recipient mice by tail vein injection. The recipient mice were then i.v. infected with $1 \times 10^6$ PFU of LCMV-Armstrong, and 48 to 60 hrs later, the recipient spleens were harvested and treated with 100 U/ml Collagenase II (Life Technologies) at 37 °C for 20 min to maximize cell recovery, and CTV dilution was tracked on CD45.2[+]GFP[+]CD8[+] cells. For characterization of effector CD8[+] T cell differentiation at peak response, $2 \times 10^4$ Vα2[+] P14 CD8[+] T cells were adoptively transferred into CD45.1[+] B6.SJL recipient mice, followed by i.p. infection with $2 \times 10^5$ PFU of LCMV-Armstrong. Eight days later, recipient spleens were harvested, and CD45.2[+]GFP[+]CD8[+] T cells were enumerated and analyzed for poly-cytokine production.

**Immunoprecipitation and Immunoblotting.** The cDNA coding N-terminus FLAG-tagged Tcf1 in the Mig-R1 retroviral vector was described[57]. The expression plasmid was transfected into 293 T cells using Lipofectamine 2000 (Life Technologies), and 24 hrs later, cell lysates were extracted and incubated overnight with 4 µl of anti-Tcf1 serum, followed by 2-hr incubation with Dynabeads Protein G (Life Technologies). After proper washing, the immunoprecipitated samples were immunoblotted with anti-FLAG antibody (clone M2, Cat. No. F3165, Millipore/Sigma-Aldrich). The cell lysates were probed with anti-FLAG to detect input proteins. For validation of protein expression changes, cell lysates were extracted from naïve WT and dKO GFP[+]CD8[+] T cells, and immunoblotted with the following antibodies: anti-Tcf1 (C46C7, Cat. No. 2206 S, Cell Signaling Technology), anti-c-MYB polyclonal antibody (Cat. No. 17800-1-AP, Proteintech), anti-CLL5 (R6G9, Cat. No. 14-7085-82, ThermoFisher Scientific), and anti-β-actin (I-19, Santa Cruz Biotechnology). The primary antibodies were used at 1:500 to 1:2000 dilution following the manufacturers' manuals.

**Statistical analysis.** The statistical significance for the multiomics analyses was assessed using the processing algorithms. Specifically, Cuffdiff was used for RNA-seq, MACS2 for Tcf1 ChIP-seq and DNase-seq, and SICER for H3K27ac ChIP-seq. The statistical significance of differential hubs was assessed using Wilcoxon signed rank test, and that associated with gene pathway and ontology analysis was assessed by DAVID and GSEA. For comparisons between two sets of data points in boxplots, one-sided Mann–Whitney $U$ test was used. For enrichment analysis, one-sided binomial test and one-sided Poisson test were used. For comparison between a contingency table, one-sided Fisher's exact test and Chi-squared test were used.

**Reporting summary.** Further information on research design is available in the Nature Research Reporting Summary linked to this article.

## Data availability
The high-throughput sequencing data generated in this study, including Hi-C, DNase-seq, RNA-seq, Tcf1 and H3K27ac ChIP-seq data, have been deposited at the Gene Expression Omnibus under primary accession number GSE164713. Source data related to Figs. 1–8 are provided as Supplementary Data 1–6, and source data for Figs. 6c and 9 are provided as a Source Data file. Source data are provided with this paper.

## Code availability
Source code for identification of hubs of differential chromatin interactions of is freely accessible to the public at: https://github.com/lux563624348/HiC_Hubs/tree/0.0.7.

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

## Acknowledgements
We thank the Flow Cytometry Core facilities at the Center for Discovery and Innovation, Hackensack University Medical Center (M. Poulos and W. Cao) and at the University of Iowa (J. Fishbaugh, H. Vignes and G. Rasmussen) for cell sorting. This study is supported in-part by grants from the NIH (AI121080 and AI139874 to H.-H.X. and W.P., AI112579 to H.-H.X.) and the Veteran Affairs BLR&D Merit Review Program (BX002903) to H.-H.X.

## Author contributions
Q.S. performed the experiments, with assistance from X.C. and K.G.; X.L. analyzed all the high-throughput sequencing data, with assistance from S.Z and Z.Z.; W.P. and H.H.X. conceived the project, supervised the study, and wrote the paper.

## Competing interests
The authors declare no competing interests.
