## [Peer Review File · Nature Communications]

REVIEWER COMMENTS

Reviewer #1 (Remarks to the Author):

In the manuscript entitled "Tcf1 and Lef1 orchestrate genomic architecture to supervise mature CD8+ T cell identity" Shan et al. utilize Hi-C together with other high-throughput sequencing techniques to determine the role of Tcf1/Lef1 TFs in supervising mature CD8 T cell identity via organizing genomic architecture and facilitating promoter-enhancer/silencer interactions. First the authors showed that Tcf1/Lef1 impacts genomic structure on multiple scales and maintains an accessible chromatin state and super enhancer activity. By comparing the gene expression profiles in WT and dKO CD8 T cells, the authors further revealed Tcf1/Lef1 TFs are necessary to suppress aberrant expression of non-T lineage and other T cell subset-associated genes. Lastly, the authors perform analyses to document an association between gene expression changes connected to SE activity and the chromatin interactions coupled to Tcf1/Lef1 binding. Collectively these data extend our understanding of the mechanisms that maintain naïve CD8 T cells in a quiescent state, however the study falls short in describing the overall contribution of Tcf1 and Lef1 in regulating the chromatin architecture of T cells as they undergo a natural reprogramming during an immune response. The general body of work is very compelling, but there are a few outstanding questions listed below, that will strengthen the claims of the manuscript once they are addressed.

1. The authors possess the tools to describe the overall chromatin architecture changes that occur during T cell effector differentiation. This analysis must be done in order to interpret the chromatin architecture changes that are truly associated with the quiescent state of naïve CD8 T cells. Furthermore, by documenting the overall change that occurs with a naïve CD8 T cell exits its quiescent state, the authors will then be able to fully resolve the overall contribution of Tcf1 and Lef1 in the regulating the chromatin architecture of naïve CD8 T cells.

2. In Tcf1/Lef1 TFs-ko CD8 T cells, the expression of effector T cell signature genes are upregulated. What is the physiological relevance of this? For example, do the naïve T cells lacking Tcf1/Lef1 TFs have a stronger tonic TCR signaling, or a lower threshold for activation such that a lower antigen quantity is now required to initiate the effector program compared to WT CD8 T cells?

3. The authors defined Motif+ and Motif- Tcf1 peaks in their CHIP-seq data as direct and indirect binding sites. Is there way to describe the overall contribution of the direct binding of Tcf1 to the effect observed in dKO CD8 T cells? Does the indirect effect also depend on the direct DNA-binding of Tcf1? Does perturbation of DNA-binding motifs results in the same changes in genomic organization or chromatin modeling as the TF ko?

Reviewer #2 (Remarks to the Author):

Shan and Li et al. provide an extremely thorough and valuable account of the impact of Tcf1 on post-thymic CD8 T cells, connecting Tcf1 binding with multiple indices of chromatin structure at and around the sites of binding, and at and around the genes affected by loss of Tcf1. Although this in vivo model does not allow identification of the earliest events when the cells lose Tcf1, the authors scrupulously test each effect against the presence or absence of Tcf1 binding at the sites involved, and they also distinguish between sites where Tcf1 appears to be bound directly and sites where the weakness of the motif suggests that it is bound indirectly. The accompanying datasets, especially the tables of differentially regulated genes and signature genes, are filled with valuable results presented in a very useful and lucid way. The context and the interpretations are very well presented in the introduction and discussion. This work will be extremely valuable to anyone interested in rigorous cause-effect analyses of transcription factor actions, and because Tcf1 is so important for T cell development and function, it will be valuable to most readers with an interest in T cell molecular biology.

Notable results, in a somewhat different order from the way they are presented, include these: (1) the impact of Tcf1 deletion after thymic egress is much more limited and different from the impact

if it is deleted during CD4/CD8 lineage separation; (2) the sets of target genes, positively and negatively regulated by Tcf1, include very strongly affected Myb and Foxp3; (3) whereas Tcf1 binding is often associated with chromatin accessibility increases, many of these sites appear to mediate negative transcriptional regulation; (4) loss of Tcf1 causes not only a shift from primarily "naive CD8" gene expression to "effector CD8" gene expression, as expected, but also striking gains in expression of genes associated with other developmental states. Particularly strong upregulation of genes associated with Treg fate could be due to the massive increase in Foxp3 expression in these cells, although this is very lightly noted in the text. In addition, gene sets associated with nonlymphoid (DC, mono, Gr) cells are seen to be upregulated, and there is even slight upregulation of some highly lineage-specific B-cell genes including the Pax5 target Cd19 and the EBF1 target Cd79a.

(5) Finally, as a technical feature, the authors have generated a new anti-Tcf1 antiserum which appears to make possible Tcf1 ChIP-seq with high sensitivity. If made available to others, this should become a vital reagent for the field.

Some aspects of presentation could be clearer. Addressing these points would help the reader through the logic of the Results section, which is presented in a fairly dry way with only occasional reference to the biology and in a slightly unexpected order.

1. The Results begin with description of the Tcf1 ChIP-seq data, but none is actually presented to the reader until Fig. 4. Not until Fig. 4 can the reader see the quality of the peaks being detected, their excellent signal:noise ratios, and the elimination of these peaks in the KO samples. This seems strange, and it would be very helpful to add an example pair of tracks around some representative loci in Fig. 1.

2. Fig. 3 provides gene expression data only in terms of relative z-scores, but some of these effects involve bare increases of barely detectable trace signals to slightly larger trace signals (e.g. for Cd19) while others, like the effect on Treg "master regulator" Foxp3, are hugely significant, not only in statistical difference but also in the level achieved. Anyone interested in the impact of Tcf1 loss on the biological state of the cells would care about these absolute differences. The data are well presented in the Supplementary Tables, but some of the biologically notable effects should be shown within the figure panels as RNA-seq tracks so that their magnitudes can be appreciated. (It is actually surprising not to see RNA-seq data included.)

3. The text explaining Fig. 3 and Fig. S5 (in lines 253-264) may be a little too brief in explaining what these "signature gene sets" mean when they are up-regulated in the KO.

(a) Readers may assume that these are highly specific indicators for a unique, alternative developmental path, but it may be more helpful to call them "characteristically enriched in" or "associated with" cells of a given alternative lineage. Some genes in the non-T cell-type signature gene sets appear to be truly lineage-specific, e.g. the B cell genes Cd19 and Cd79a, but not all of them are this clear. For example, the essential T-cell regulatory gene Gata3 is included in the "Pan-NK signature". In the full ImmGen database, Cish, described as "Treg signature", is highly upregulated in ILC2 cells as well, and Cx3cr1, described as "effector CD8 signature", is almost equally high in certain monocytes and much higher, in fact, in microglia.

(b) Also, although the signature gene set as a whole may be quite biased to a non-T cell type, the particular members of that gene set that are up-regulated in the Tcf1 KO cells are often expressed at one time or another in T lineage cells (spot checks show many of them in thymocytes, etc.). It would be helpful if the authors slightly expanded the description of the signature gene sets in the text to clarify that these particular genes do not mean that the cells are actually transforming into DC or B cells. Otherwise, the reader would look for more detailed flow cytometry to characterize the phenotype of the cells more completely.

4. Fig. 2 makes a strong distinction between Motif+ and Motif- Tcf1 binding peaks, with the majority of the binding peaks lying in an indeterminate motif score range. After encountering the Tcf1-regulated genes, the question is whether this distinction makes a difference to the effect of the Tcf1 binding on the response of the target genes. Although this is discussed on p. 15, Figure 4a (right panel) shows an indistinct kind of heatmap view where all the Tcf1 binding sites appear to be associated with some kind of Tcf1 motif, but little pattern is detectable. Maybe the Tcf1 motif strength differences are overwhelmed by the divide between Tcf1-bound and Tcf1-unbound sites.

If the Tcf1-bound sites are separated out from the unbound sites, can one then see a difference between Tcf/Lef motif strength in C2 and C4, or any relation to the response of the target gene?

5. On p. 17, the text makes it seem that the effects of Tcf1 KO on positively regulated SEs are just as strong in the opposite direction on negatively regulated SEs. But H3K27ac is clearly lost from specific peaks within the positively regulated SEs in the KO (Ccr7 and Inpp4b loci), while in contrast, the increase in H3K27ac in the negatively regulated ones (around Cish and around Cx3cr1) seems to be just a global increase in background H3K27ac between the peaks. At first glance, the KO samples just appear to have a worse signal-to-noise ratio. Is this the general way that the Tcf1/Lef1 repressed loci all over the genome react? Perhaps the specificity of the effect on the repressed loci would show more clearly if the panels in Fig. 5f were more zoomed-out.

5. Minor:

(a) in Fig. 6, please clarify where panel b ends and panel c begins. The labeling suggests that the middle hub map is still b, but the red/blue arrow features are only described under the legend for c.

(b) In Fig. S4, please explain what the small dots are. Are they projections of the large dots onto each 2D plane?

(c) The biologically interested reader would probably welcome some direct comments about the strong effects of Tcf1 KO on powerful regulatory genes in these CD8 cells, like Myb (greatly down in the KO), Foxp3 (greatly up), Maf (up), and Eomes (down).

(d) Please clarify whether the new Tcf1 antiserum will be available for others to try using, and where a more complete characterization of its properties relative to commercial antibodies may be found.

Reviewer #3 (Remarks to the Author):

Review of manuscript NCOMMS-21-06081-T "Tcf1 and Lef1 orchestrate genomic architecture to supervise mature CD8+ T cell identity" by Shan, Li et. al..

The authors explore the role of Tcf1 and Lef1 in the maintenance of CD8+ T cell identity with a focus on the integrity of chromatin architecture and chromatin accessibility or gene expression after hCD2-Cre mediated deletion of these two transcription factors in mature peripheral T cells. The authors present a new Tcf1 ChIP and use the determined Tcf1 peaks to systematically assess several global analyses including chromatin topology / Hi-C Seq, chromatin accessibility / DNase seq, superenhancer formation as well as gene expression by discriminating Tcf1+ peak and Tcf1-Motif+ containing from Tcf1- genomic regions and WT/DKO comparisons. Besides providing a profound analysis of the different levels at which Tcf1/Lef1 regulate chromatin accessibility and gene expression or prevent cell-type inappropriate gene expression the most intriguing results are the Tcf1-dependent changes in chromatin topology.

Major point 1. The authors study primary CD8+ T cells and have established an elegant system of conditional deletion (plus identification of deleted cells) to study Tcf1 and Lef1 function, but they should also describe or at least mention some of the relevant phenotypes of these DKO CD8+ T cells. If the cells lose their identity, one expects them to show functional impairments for example in killing of target cells, changed cytokine/chemokine production, they should inappropriately adopt certain effector phenotypes, undergo exhaustion, have altered survival and express protein markers of other cell types (CD19??)?

Major point 2. The authors generate new polyclonal anti-TCF1 specific antibodies for ChIP use, but they do not show a validation of specificity.

Major point 3. For Fig. S3a, the authors conclude that Tcf1 peaks are absent from CTCF marked boundaries of TADs. However, Tcf1 peaks appear enriched upstream and downstream of the boundaries (approx. 100kb distance). How is this enrichment explained?

Moreover, I do not follow how the authors describe the localization of Tcf1 peaks within the displayed TADs for example in Fig. 1i... "a chromatin loop linking *Cyct* and *Prkra* gene loci in WT cells with a Tcf1 peak at the anchor proximal to *Cyct*...". I rather see the peak positioned at or close to the first exon of *Rbm45*, and in general (by judging the displayed examples), I would rather see Tcf1 peaks proximal to the genes than at specific anchor points of loops. Please clarify how loop anchors and the presence or absence of Tcf1 peaks within them are defined.

Major point 4. A criticism for the chromatin topology aspects of this study is that although the authors globally map Tcf1 binding sites by ChIP, define them to be enriched in "transcriptionally more active regions" and in regions with higher TAD scores, they only show diminished TADs in DKO CD8+ T cells for genes that are not prototypic targets of Tcf1/Lef1. Can the authors include examples of a bona fide target (maybe *Prdm1* or even *CD4* etc.) that have specific Tcf1 binding sites and important functions in T cells and demonstrate that the globally observed effects can also be seen on the well-established and relevant target -- or is the conclusion that the "structural role" of Tcf1 applies rather outside of the set of already established target genes?

Major point 5. The authors generated mouse mutants with ablation of two binding sites in *Prdm1* that are associated with unchanged or reduced ChrAcc at these positions in DKO T cells and either reduce or enhance *Prdm1* expression. Using these mice they could directly test the suggested interpretation that Tcf1 acts as a pioneer factor via bending of chromatin, increasing accessibility locally and allowing looping and altered chromatin topology, which through interaction with other transcription factors can then have very different impacts on the gene expression.

Point-by-point response to REVIEWER COMMENTS

Reviewer #1 (Remarks to the Author):

In the manuscript entitled "Tcf1 and Lef1 orchestrate genomic architecture to supervise mature CD8+ T cell identity" Shan et al. utilize Hi-C together with other high-throughput sequencing techniques to determine the role of Tcf1/Lef1 TFs in supervising mature CD8 T cell identity via organizing genomic architecture and facilitating promoter-enhancer/silencer interactions. First the authors showed that Tcf1/Lef1 impacts genomic structure on multiple scales and maintains an accessible chromatin state and super enhancer activity. By comparing the gene expression profiles in WT and dKO CD8 T cells, the authors further revealed Tcf1/Lef1 TFs are necessary to suppress aberrant expression of non-T lineage and other T cell subset-associated genes. Lastly, the authors perform analyses to document an association between gene expression changes connected to SE activity and the chromatin interactions coupled to Tcf1/Lef1 binding. Collectively these data extend our understanding of the mechanisms that maintain naïve CD8 T cells in a quiescent state, however the study falls short in describing the overall contribution of Tcf1 and Lef1 in regulating the chromatin architecture of T cells as they undergo a natural reprogramming during an immune response. The general body of work is very compelling, but there are a few outstanding questions listed below, that will strengthen the claims of the manuscript once they are addressed.

We thank the reviewer for many positive comments on this work.

1. The authors possess the tools to describe the overall chromatin architecture changes that occur during T cell effector differentiation. This analysis must be done in order to interpret the chromatin architecture changes that are truly associated with the quiescent state of naïve CD8 T cells. Furthermore, by documenting the overall change that occurs with a naïve CD8 T cell exits its quiescent state, the authors will then be able to fully resolve the overall contribution of Tcf1 and Lef1 in the regulating the chromatin architecture of naïve CD8 T cells.

We understand the reviewer's reasoning in recommending analyses of chromatin architecture in differentiated effector CD8+ T cells. Unlike transcription factors such as Runx3 which remains stably expressed during CD8+ T cell response to infection (1, 2), Tcf1 and Lef1 expression are greatly diminished in terminally differentiated effector CD8+ T cells in the context of acute infections (3, 4). In fact, ectopic Tcf1 expression impedes effector differentiation (3, 5). Based on these observations, we posit that Tcf1/Lef1-dependent chromatin architecture in a naïve CD8+ T cell have to be extensively reorganized to facilitate its differentiation to an effector CD8+ T cell. Because of the potent downregulation of Tcf1 and Lef1 during "a natural reprogramming during an immune response", we further reason that the contribution of Tcf1 and Lef1 to chromatin re-organization is rather limited compared with TCR-mobilized transcription factors in the AP1, NFAT and NF- κ B families and TCR-induced Myc and Egr transcription factors. The validity of this view is supported by our analyses of differentiation of WT and dKO naïve CD8+ T cells into effector cells (detailed below, data in Fig. 8 and S8). While there is no underestimation of the importance to understand chromatin architecture of effector T cells, we respectfully suggest that systematic Hi-C analysis of effector T cells is beyond the scope of this work and requires dedicated studies in the future.

The reviewer's concerns, in conjunction with point #2, are valid with regard to whether the extensive alternations of chromatin architecture observed in Tcf1/Lef1-deficient naïve CD8+ T cells are biologically important. Our molecular analyses revealed that besides maintaining the quiescent state of naïve CD8+ T cells, Tcf1 and Lef1 are essential for positive regulation of Ccr7, Eomes, and Myb, and negative regulation of Foxp3 and non-T lineage signature genes. We

validated protein expression of key Tcf1/Lef1 target genes and examined their associated biological functions, and made the following observations:

1. Reduced CCR7 protein expression in dKO CD8⁺ T cells is associated with their diminished capacity of homing to lymph nodes (Fig. 8a, b);
2. Decreased EOMES protein expression in dKO CD8⁺ T cells is associated with reduced frequency and numbers of CD44 high CD122⁺ memory-phenotype CD8⁺ T cells in uninfected mice (Fig. 8c, d);
3. We validated diminished MYB and elevated FOXP3 protein expression in dKO CD8⁺ T cells. MYB supports effector T cell expansion and polyfunctionality (6), and Foxp3 has a broad transcription repressor function (7). Upon tested in vivo, dKO CD8⁺ T cells exhibited reduced proliferative capacity within 60 hrs of activation, leading to profound reduction in effector CD8⁺ T cells at the peak response to viral infection (Fig. 8e-k). dKO effector cells also showed diminished polyfunctionality in terms of cytokine production (Fig. S8e,f), These defects were in fact a phenocopy of MYB ablation in CD8⁺ T cells (6).

These data are now described on pages 23-25, lines 493-546. Collectively, these findings demonstrate the biological importance of Tcf1/Lef1 in naïve CD8⁺ T cells in supporting T cell program and repressing lineage-inappropriate genes by employing multifaceted mechanisms. These concerted actions by Tcf1/Lef1 are critical for ensure efficient and proper differentiation into cytotoxic effector cells.

2. In Tcf1/Lef1 TFs-ko CD8 T cells, the expression of effector T cell signature genes are upregulated. What is the physiological relevance of this? For example, do the naïve T cells lacking Tcf1/Lef1 TFs have a stronger tonic TCR signaling, or a lower threshold for activation such that a lower antigen quantity is now required to initiate the effector program compared to WT CD8 T cells?

This is a valid question and we set out to test these ideas. We first validated that granzyme B and CCL5 proteins were detected at increased levels in dKO CD8⁺ T cells (Fig. S8a,b). We then tested if these changes made dKO CD8⁺ T cells prone to activation. We used titrating amounts of plate-bound CD3 and soluble CD28 to stimulate CD8⁺ T cell ex vivo and measured interferon-gamma production and granzyme B induction. As shown in Fig. S8d, WT and dKO CD8⁺ T cells show similar sensitivity to different doses of TCR stimulation, suggesting that dKO CD8⁺ T cells did not exhibit a lower threshold for activation. Coupled with our mechanistic studies of Tcf1/Lef1-mediated target gene regulation [for example, negative regulation of chromatin accessibility at the *Gzmb* TSS (Fig. 4b), and restraint of chromatin interaction at the *Ccl* loci (Fig. 7d)], we deduce that the increased basal expression of effector program in naïve dKO CD8⁺ T cells was a result of direct regulation at the transcriptional level, while the contribution of aberrant TCR signaling may not be as significant.

In assessing the functional impact of Tcf1/Lef1 deficiency in vivo, as detailed in response to point #1, the intact expression of Tcf1/Lef1 in CD8⁺ T cells is critical for the proliferative capacity and polyfunctionality during differentiation of naïve to effector CD8⁺ cells. These biological requirements can be, at least in part, ascribed to Tcf1/Lef1-mediated positive regulation of Myb and negative regulation of Foxp3.

3. The authors defined Motif+ and Motif- Tcf1 peaks in their CHIP-seq data as direct and indirect binding sites. Is there way to describe the overall contribution of the direct binding of Tcf1 to the effect observed in dKO CD8 T cells? Does the indirect effect also depend on the direct DNA-binding of Tcf1? Does perturbation of DNA-binding motifs result in the same changes in genomic organization or chromatin modeling as the TF ko?

With regard to the “overall contribution of the direct binding of Tcf1 to the effect observed in dKO CD8 T cells”, we added a summarizing paragraph in the discussion on page 28, lines 610-618, as follows: “As evident in our systematic molecular analyses, Tcf1 binding events are associated with distinct regulatory effects in a context-dependent manner, such as promoting or disengaging chromatin interaction, increasing or reducing chromatin accessibility, and activating or repressing target gene expression. By use of position-weight matrix in motif analysis, we made the distinction between Tcf1 direct vs. indirect binding events and assessed their relative contribution to each regulatory mechanism. At its direct binding sites, Tcf1 exhibited clearly distinguishable preference for promoting chromatin interactions (Fig. 1i-k) and maintaining chromatin at an open status (Fig. 2b,c). Such distinction predicts likelihood of a preferred biological outcome, but should not be interpreted in absolute terms.”

To further clarify on this point, we added a panel in Fig. 1k to demonstrate a stronger contribution of Tcf1 direct binding to promoting formation of chromatin loops. In Fig. 2b, we added statistical values to the comparisons between Motif+ and Motif- Tcf1 binding peaks in regulating chromatin accessibility.

With regard to the other point, perturbing Tcf1 DNA binding motif is an excellent idea, and we are interested in the same question as the reviewer. As illustrated in the diagram below, the HMG DNA binding domain starts from the 304th residue in full length Tcf1 protein.

We attempted to address this question by inserting a stop codon at corresponding exon in mouse germline. Upon analysis of the resulting mouse strain, the Tcf1-HMG truncated protein was not stable, and resulted in a null mutant as shown below.

[Redacted]

This was reported previously in J. Immunol. (8). This observation indicates that a more subtle approach, such as specific mutation of DNA-contacting residues in Tcf1, is necessary for future dedicated investigation.

Reviewer #2 (Remarks to the Author):

Shan and Li et al. provide an extremely thorough and valuable account of the impact of Tcf1 on post-thymic CD8 T cells, connecting Tcf1 binding with multiple indices of chromatin structure at and around the sites of binding, and at and around the genes affected by loss of Tcf1. Although this in vivo model does not allow identification of the earliest events when the cells lose Tcf1, the authors scrupulously test each effect against the presence or absence of Tcf1 binding at the sites involved, and they also distinguish between sites where Tcf1 appears to be bound directly and sites where the weakness of the motif suggests that it is bound indirectly. The accompanying datasets, especially the tables of differentially regulated genes and signature genes, are filled with valuable results presented in a very useful and lucid way. The context and the interpretations are very well presented in the introduction and discussion. This work will be extremely valuable to anyone interested in rigorous cause-effect analyses of transcription factor actions, and because Tcf1 is so important for T cell development and function, it will be valuable to most readers with an interest in T cell molecular biology.

We thank the reviewer for many positive comments on this work.

Notable results, in a somewhat different order from the way they are presented, include these: (1) the impact of Tcf1 deletion after thymic egress is much more limited and different from the impact if it is deleted during CD4/CD8 lineage separation; (2) the sets of target genes, positively and negatively regulated by Tcf1, include very strongly affected Myb and Foxp3; (3) whereas Tcf1 binding is often associated with chromatin accessibility increases, many of these sites appear to mediate negative transcriptional regulation; (4) loss of Tcf1 causes not only a shift from primarily “naïve CD8” gene expression to “effector CD8” gene expression, as expected, but also striking gains in expression of genes associated with other developmental states. Particularly strong upregulation of genes associated with Treg fate could be due to the massive increase in Foxp3 expression in these cells, although this is very lightly noted in the text. In addition, gene sets associated with nonlymphoid (DC, mono, Gr) cells are seen to be upregulated, and there is even slight upregulation of some highly lineage-specific B-cell genes including the Pax5 target Cd19 and the EBF1 target Cd79a. (5) Finally, as a technical feature, the authors have generated a new anti-Tcf1 antiserum which appears to make possible Tcf1 ChIP-seq with high sensitivity. If made available to others, this should become a vital reagent for the field.

We thank the reviewer for the nice summary of our key findings. Coupled with responses to the other two reviewers, we enriched our discussion on a few target genes with strong relevance to T cell biology.

Some aspects of presentation could be clearer. Addressing these points would help the reader through the logic of the Results section, which is presented in a fairly dry way with only occasional reference to the biology and in a slightly unexpected order.

1. The Results begin with description of the Tcf1 ChIP-seq data, but none is actually presented to the reader until Fig. 4. Not until Fig. 4 can the reader see the quality of the peaks being detected, their excellent signal:noise ratios, and the elimination of these peaks in the KO samples. This seems strange, and it would be very helpful to add an example pair of tracks around some representative loci in Fig. 1.

As requested, we have added sample tracks as Fig. 1a, where known Tcf1 targets, Cd4 and Tcf7 gene itself, are displayed. The description was updated on page 5, lines 108-112.

2. Fig. 3 provides gene expression data only in terms of relative z-scores, but some of these

effects involve bare increases of barely detectable trace signals to slightly larger trace signals (e.g. for Cd19) while others, like the effect on Treg “master regulator” Foxp3, are hugely significant, not only in statistical difference but also in the level achieved. Anyone interested in the impact of Tcf1 loss on the biological state of the cells would care about these absolute differences. The data are well presented in the Supplementary Tables, but some of the biologically notable effects should be shown within the figure panels as RNA-seq tracks so that their magnitudes can be appreciated. (It is actually surprising not to see RNA-seq data included.)

We agree with the reviewer on that the actual levels of a gene transcript are another important aspect besides fold changes. We chose to use a Supplemental Table to present such data in place of actual tracks for the following technical reasons:

1. When presenting long-range chromatin interactions and super enhancers in Figs. 5-7, the regions of interest usually encompass multiple genes with varied expression levels. Compared with ATAC-seq and H3K27ac ChIP-seq, the RNA-seq signals were in much wider ranges, even for neighboring genes. It was therefore difficult to highlight the expression changes for a DEG while keeping data of neighboring genes informative.
2. To highlight ChrAcc changes in Fig. 4, it was necessary to exhibit longer intergenic region for small genes (such as Gzmb in Fig. 4b), or partial gene structure for large genes (such as Pax5, covering almost 200k region, in Fig. 4d). In these scenarios, the small genes were too condensed, while the transcripts from partial exons did not convey all the information on the whole gene transcription activity.

For these considerations and to keep all display panels consistent in this manuscript, we now show the RNA-seq tracks of select genes in separate panels (all in Fig. S5). The gene selection was based on expanded description for lineage-enriched genes as detailed in response to comment #3, and on biological relevance. In these panels, we displayed all the exons in a fully legible range, adjusted y-axis heights to reflect absolute transcript levels (such as Ccr7 vs Cd19 in Fig. S5a) and the dynamic changes between WT and dKO CD8+ T cells (such as Ccl5 in Fig. S5a, Gzmb and Foxp3 in Fig. S5b).

In addition, for select biologically important genes, we validated the expression changes on protein levels (such as CCR7, MYB, EOMES, and FOXP3 in Fig. 8), and performed functional analyses, as requested by other reviewers.

3. The text explaining Fig. 3 and Fig. S5 (in lines 253-264) may be a little too brief in explaining what these “signature gene sets” mean when they are up-regulated in the KO.

(a) Readers may assume that these are highly specific indicators for a unique, alternative developmental path, but it may be more helpful to call them “characteristically enriched in” or “associated with” cells of a given alternative lineage. Some genes in the non-T cell-type signature gene sets appear to be truly lineage-specific, e.g. the B cell genes Cd19 and Cd79a, but not all of them are this clear. For example, the essential T-cell regulatory gene Gata3 is included in the “Pan-NK signature”. In the full ImmGen database, Cish, described as “Treg signature”, is highly upregulated in ILC2 cells as well, and Cx3cr1, described as “effector CD8 signature”, is almost equally high in certain monocytes and much higher, in fact, in microglia.

We agree with the reviewer’s assessment about these genes. Follow the recommendation, now we call these genes as “lineage-enriched genes (LEGs)” instead of “signature genes”, which more accurately reflect how these genes were defined. This has been changed throughout the manuscript including text, Fig. 3 and S6, figure legends, methods and Table S1-S3. In the text we clarified on the definition of LEGs and the underlying reasoning (page 12, lines 262-263, and page 13, lines 268-272).

(b) Also, although the signature gene set as a whole may be quite biased to a non-T cell type, the particular members of that gene set that are up-regulated in the Tcf1 KO cells are often expressed at one time or another in T lineage cells (spot checks show many of them in thymocytes, etc.). It would be helpful if the authors slightly expanded the description of the signature gene sets in the text to clarify that these particular genes do not mean that the cells are actually transforming into DC or B cells. Otherwise, the reader would look for more detailed flow cytometry to characterize the phenotype of the cells more completely.

This is indeed a very important point that we should have highlighted, and we thank the reviewer for the insightful input. We expanded the description of LEGs by highlighting a few genes with demonstrated function in different immune cell lineages (page 13, lines 274-281; and page 14, lines 292-296). We added a note to acknowledge that “It should be noted, however, the increased transcripts of non-T or non-cytotoxic lineage genes in dKO CD8+ T cells do not mean that Tcf1/Lef1-deficient CD8+ T cells were transformed into other cell types such as B cells, DCs or Treg cells” (page 14, lines 300-304), so as to better inform the readers. Additional functional analyses of dKO CD8 T cells, as requested by other referees, support this interpretation, because dKO CD8+ T cells retained the capacity of inducing cytotoxic cytokines upon activation (please find more details in Fig. 8, and related description on pages 23-25).

4. Fig. 2 makes a strong distinction between Motif+ and Motif- Tcf1 binding peaks, with the majority of the binding peaks lying in an indeterminate motif score range. After encountering the Tcf1-regulated genes, the question is whether this distinction makes a difference to the effect of the Tcf1 binding on the response of the target genes. Although this is discussed on p. 15, Figure 4a (right panel) shows an indistinct kind of heatmap view where all the Tcf1 binding sites appear to be associated with some kind of Tcf1 motif, but little pattern is detectable. Maybe the Tcf1 motif strength differences are overwhelmed by the divide between Tcf1-bound and Tcf1-unbound sites. If the Tcf1-bound sites are separated out from the unbound sites, can one then see a difference between Tcf1/Lef motif strength in C2 and C4, or any relation to the response of the target gene?

The distinction between Motif+ vs Motif- Tcf1 peaks was meant to distinguish if Tcf1 direct binding events were associated with a preferred functional output. This was indeed the case, and we found that Tcf1 direct binding were linked to promoting chromatin interaction (Fig. 1i-k) and keeping ChrAcc sites at an open state (Fig. 2c). We clarified the rationale in Results on page 6, lines 114-117 and 120-123, and summarized this point in Discussion (page 28, lines 610-618).

We appreciate the reviewer’s suggestion on separating Tcf1-bound and unbound sites to discern a detectable pattern. Because the use of position weight matrix in motif analysis provides a quantifiable “motif score”, we have now updated the motif panel, using color-coded horizontal lines to mark the category of motif strength of Tcf1 peaks associated with Diff ChrAcc sites (Fig. 4a, far right column).

We further examined the C2 and C4 Diff. ChrAcc sites, and found that all of 79 C2 sites and 98 out of 108 C4 sites overlapped with high confidence Tcf1 peaks. By applying the motif scores to these sites, we observed that C2 cluster showed modest enrichment with Motif+ Tcf1 direct binding events than C4 ($p = 0.034$ by Chi-square test). We have now included these data in Fig. 4g and updated the data description on page 16, lines 352-354.

5. On p. 17, the text makes it seem that the effects of Tcf1 KO on positively regulated SEs are just as strong in the opposite direction on negatively regulated SEs. But H3K27ac is clearly lost from specific peaks within the positively regulated SEs in the KO (Ccr7 and Inpp4b loci), while

in contrast, the increase in H3K27ac in the negatively regulated ones (around Cish and around Cx3cr1) seems to be just a global increase in background H3K27ac between the peaks. At first glance, the KO samples just appear to have a worse signal-to-noise ratio. Is this the general way that the Tcf1/Lef1 repressed loci all over the genome react? Perhaps the specificity of the effect on the repressed loci would show more clearly if the panels in Fig. 5f were more zoomed-out.

The reviewer proposed an interesting scenario on the different behavior of H3K27ac on WT- and dKO-prepotent SEs. To test this idea, we first extracted the total H3K27ac reads (including background reads) at the 174 WT- and 163 dKO-prepotent SEs. We then obtained reads at H3K27ac islands, which were called by the SICER algorithm as H3K27ac antibody-enriched genomic regions over IgG ChIP-derived signals. As summarized in the table below, the signal to noise ratios (island-filtered reads over total reads) were similar between WT- and dKO-prepotent SEs, suggesting that the differential SE signals are not due to different noise levels in either group.

Diff. SE	Total reads	Island-filtered reads	Signal/Noise ratio
WT-prepotent Ses (174)	4,517,610	4,210,134	0.9319
dKO-prepotent SEs (163)	3,773,000	3,519,044	0.9327

Another point of clarification is that although an SE is marked with a continuous bar on the genome browser, the individual H3K27ac islands that constitute an SE are discontinuous, as exemplified at the *Cish* locus shown below for the SE and K27ac marking. When defining an H3K27ac island, the SICER algorithm takes continuous signals from neighboring windows into consideration; as such, the H3K27ac islands, as shown below the SE track, include not only apparent peaks but also regions flanking the peaks and regions between the apparent peaks. These might be the underlying reasons for the perceived higher background noise in some SEs, especially the few dKO-prepotent ones in Fig. 5f. We further clarified this point in Methods and cited the Signal/Noise ratio in the Table above (page 52, line 1146; page 53, lines 1155-1159).

6. Minor:

(a) in Fig. 6, please clarify where panel b ends and panel c begins. The labeling suggests that the middle hub map is still b, but the red/blue arrow features are only described under the legend for c.

As requested, we added a frame to panels b and c to mark the boundaries.

(b) In Fig. S4, please explain what the small dots are. Are they projections of the large dots onto each 2D plane?

The reviewer is correct. We have clarified in the figure legends to Fig. S4, as the follows: “The small dots are projections of each replicate to respective 2D planes.”

(c) The biologically interested reader would probably welcome some direct comments about the strong effects of Tcf1 KO on powerful regulatory genes in these CD8 cells, like Myb (greatly down in the KO), Foxp3 (greatly up), Maf (up), and Eomes (down).

Motivated by this comment along with input from other reviewers, we have now validated protein expression of EOMES, MYB and FOXP3 in dKO CD8+ T cells, and analyzed their link to biological changes in the dKO CD8+ T cells, such as accumulation of memory-phenotype CD8+ T cells in uninfected mice (Fig.8c,d), and proliferative capacity in response to antigen stimulation (Fig. 8e–k). For detailed description, please refer to pages 23-25.

(d) Please clarify whether the new Tcf1 antiserum will be available for others to try using, and where a more complete characterization of its properties relative to commercial antibodies may be found.

As requested, we included data on characterization of the Tcf1 antiserum. We demonstrated its ability to detect FLAG-tagged Tcf1 (Fig. S2a) and endogenous Tcf1 proteins/isoforms in CD8+ T cells (Fig. S2b) by immunoblotting. We further demonstrated its ability to immunoprecipitate Tcf1 protein in Fig. S2c. The antiserum will be made available upon request, and this is noted in the text, page 6, line 112.

Reviewer #3 (Remarks to the Author):

The authors explore the role of Tcf1 and Lef1 in the maintenance of CD8+ T cell identity with a focus on the integrity of chromatin architecture and chromatin accessibility or gene expression after hCD2-Cre mediated deletion of these two transcription factors in mature peripheral T cells. The authors present a new Tcf1 ChIP and use the determined Tcf1 peaks to systematically assess several global analyses including chromatin topology / Hi-C Seq, chromatin accessibility / DNase seq, super enhancer formation as well as gene expression by discriminating Tcf1+ peak and Tcf1-Motif+ containing from Tcf1- genomic regions and WT/DKO comparisons. Besides providing a profound analysis of the different levels at which Tcf1/Lef1 regulate chromatin accessibility and gene expression or prevent cell-type inappropriate gene expression the most intriguing results are the Tcf1-dependent changes in chromatin topology.

We thank the reviewer for the positive comments on this work.

Major point 1. The authors study primary CD8+ T cells and have established an elegant system of conditional deletion (plus identification of deleted cells) to study Tcf1 and Lef1 function, but they should also describe or at least mention some of the relevant phenotypes of these DKO CD8+ T cells. If the cells lose their identity, one expects them to show functional impairments for example in killing of target cells, changed cytokine/chemokine production, they should inappropriately adopt certain effector phenotypes, undergo exhaustion, have altered survival and express protein markers of other cell types (CD19?)?

We agree with the reviewers that it is important to examine the phenotypic and functional changes in the dKO CD8+ T cells, in conjunction with all molecular alterations. We performed

the following studies, which are collectively shown in new Fig. 8 and S8 and described on pages 23-25. Key findings are briefly summarized below:

1. Reduced CCR7 protein expression in dKO CD8⁺ T cells is associated with their diminished capacity of homing to lymph nodes (Fig. 8a, b);
2. Decreased EOMES protein expression in dKO CD8⁺ T cells is associated with reduced frequency and numbers of CD44 high CD122⁺ memory-phenotype CD8⁺ T cells in uninfected mice (Fig. 8c, d);
3. We validated diminished MYB and elevated FOXP3 protein expression in dKO CD8⁺ T cells. MYB supports effector T cell expansion and polyfunctionality (6), and Foxp3 has a broad transcription repressor function (7). Upon tested in vivo, dKO CD8⁺ T cells exhibited reduced proliferative capacity within 60 hrs of activation, leading to profound reduction in effector CD8⁺ T cells at the peak response to viral infection (Fig. 8e-k). dKO effector CD8⁺ cells also showed diminished polyfunctionality in terms of cytokine production (Fig. S8e,f). These defects were a phenocopy of MYB ablation in CD8⁺ T cells (6).
4. We also validated increased basal expression of Granzyme B and CCL5 production in naïve dKO CD8⁺ T cells, but these changes did not result in lower threshold for activation of dKO cells (Fig. S8a, b, d).
5. dKO CD8⁺ T cells were not detectably more prone to apoptosis than WT cells (Fig. S8c).

Collectively, specific deletion of Tcf1/Lef1 in naïve CD8⁺ T cells resulted in extensive perturbation of their functionality, as a result of molecular changes in ChrAcc, super enhancer activity, and chromatin topology due to loss of Tcf1/Lef1 TFs.

Major point 2. The authors generate new polyclonal anti-TCF1 specific antibodies for ChIP use, but they do not show a validation of specificity.

As requested, we included data on characterization of the Tcf1 antiserum. We demonstrated its ability to detect FLAG-tagged Tcf1 (Fig. S2a) and endogenous Tcf1 proteins/isoforms in CD8⁺ T cells (Fig. S2b) by immunoblotting. We further demonstrated its ability to immunoprecipitate Tcf1 protein in Fig. S2c. The data are described on page 6, lines 108-109.

Major point 3. For Fig. S3a, the authors conclude that Tcf1 peaks are absent from CTCF marked boundaries of TADs. However, Tcf1 peaks appear enriched upstream and downstream of the boundaries (approx. 100 kb distance). How is this enrichment explained?

The original plot had TAD boundary in the center. As a result, the aggregated Tcf1 ChIPseq signals in TADs to the left of the boundary and those in TADs to the right of the boundary both appeared in the same plot.

To further clarify this point, we added another plot to Fig. S3a, where the left and right TAD boundaries were aligned, with TAD in the middle. In this plot, CTCF ChIPseq signals were more enriched at TAD boundaries while Tcf1 ChIPseq signals were more enriched within the TADs. Both plots now demonstrate the same point from two vantage points. The figure legend was updated accordingly.

Moreover, I do not follow how the authors describe the localization of Tcf1 peaks within the displayed TADs for example in Fig. 1i... "a chromatin loop linking Cyt and Prkra gene loci in WT cells with a Tcf1 peak at the anchor proximal to Cyt...". I rather see the peak positioned at or close to the first exon of Rbm45, and in general (by judging the displayed examples), I would rather see Tcf1 peaks proximal to the genes than at specific anchor points of loops.

Please clarify how loop anchors and the presence or absence of Tcf1 peaks within them are defined.

We thank the reviewer for the keen observation. For the referred example (now in Fig. 1j), it was our inadvertent error. The specific chromatin loop connects Tcf1-bound Rbm45 TSS with an upstream region of Prkra (as marked with dotted green lines for the interacting bins). This has been corrected in the text (page 9, lines 181-183) and figure legend (page 34, line 734).

Major point 4. A criticism for the chromatin topology aspects of this study is that although the authors globally map Tcf1 binding sites by ChIP, define them to be enriched in “transcriptionally more active regions” and in regions with higher TAD scores, they only show diminished TADs in DKO CD8+ T cells for genes that are not prototypic targets of Tcf1/Lef1. Can the authors include examples of a bona fide target (maybe Prdm1 or even CD4 etc.) that have specific Tcf1 binding sites and important functions in T cells and demonstrate that the globally observed effects can also be seen on the well-established and relevant target -- or is the conclusion that the “structural role” of Tcf1 applies rather outside of the set of already established target genes?

We understand the reviewer’s comments. Resolution of Hi-C data is one of the major limiting factors. Our Hi-C data reached a 10-kb resolution, which is on par with published Hi-C data on primary immune cells (9, 10). We recognize that this resolution remains substantially lower than DNase-seq peaks or Tcf1 peaks. For loop calling, we employed the widely used HiCCUPs algorithm in the Juicer suite (11, 12), which does not detect short-range chromatin interactions. From our Hi-C data, the shortest distance of HiCCUPs-identified chromatin loops is 70 kb. In addition, a high-confidence chromatin loop called by HiCCUPs typically does not depend on a single bin-to-bin connection (which appears as a single pixel on the diamond graph), but requires 5–20 neighboring pixels that support the interaction in the matrix (12). This point is also highlighted in the diamond graph above, where the chromatin loop connecting Rbm45 and Prkra is supported by interactions from neighboring bins.

We share the same interest as the reviewer, in defining an architectural role in regulation of known Tcf1/Lef1 target genes. The *Prdm1* and *Cd4* genes encompasses 20.4 kb and 23.5 kb, respectively. The *Prdm1* upstream silencer, as defined by DNase-seq and H3K27ac ChIPseq data (Fig. 5b, c) is 24 kb from its TSS. In all these cases, the distance between key regulatory elements and gene promoters is not large enough to be sufficiently resolved for identification of chromatin loops. As displayed in diamond graphs for the genes below, although there are discernible, isolated pixels showing a difference between WT and dKO CD8 T cells (yellow

arrows), those differences do not meet the stringent statistical criteria to be identified as differential chromatin loops.

On the other hand, we did reliably detect long range interactions at the *Myb* and *Ccl3,4,5* genes, where profound alterations in chromatin topology were observed between WT and dKO CD8⁺ T cells (Fig. 6b,c). Although these genes were not the known target genes of Tcf1/Lef1, following the recommendations of the reviewer and other referees, we were able to validate that these Tcf1/Lef1-dependent architectural changes affect their protein expression (Fig. 8e and S8b). Using an in vivo infection model, we further demonstrated that diminished *Myb* expression in the dKO CD8 T cells at least partly account for the defects in proliferative capacity and polyfunctionality of activated antigen-specific dKO CD8⁺ T cells (Fig. 8g-k, S8e,f) (6).

These new findings demonstrate the usefulness of mapping chromatin topological changes in identifying novel, biologically relevant regulatory circuits controlled by Tcf1/Lef1. Admittedly, many genes identified this way do not have known function in T cell biology at present, but they may become useful resource for future studies as our knowledge on T cells expands.

Major point 5. The authors generated mouse mutants with ablation of two binding sites in *Prdm1* that are associated with unchanged or reduced ChrAcc at these positions in DKO T cells and either reduce or enhance *Prdm1* expression. Using these mice they could directly test the suggested interpretation that Tcf1 acts as a pioneer factor via bending of chromatin, increasing accessibility locally and allowing looping and altered chromatin topology, which through interaction with other transcription factors can then have very different impacts on the gene expression.

We understand the reviewer's recommendation, which is fine idea. As discussed in our response to Major Point #4, due to resolution limitation from the current Hi-C protocol, small size of the *Prdm1* gene (20.4 kb) and the relatively short distance of the upstream regulatory element (24 kb), a definitive difference in element-promoter interaction cannot be conclusively observed between WT and Tcf1/Lef1 dKO cells. Please also refer to the diamond graph shown above. This would likely be the case for *Prdm1* element mutant cells. On pages 29-30, lines 642-657, we acknowledged this limitation, and discuss the necessity of increasing HiC resolution for more accurate assessment of short-range interactions.

On a positive note, through mapping of ChrAcc and H3K27ac state, we were able to identify a Tcf1/Lef1-dependent silencer for controlling *Prdm1* expression in naïve CD8⁺ T cells. These complementary approaches did help resolve key regulatory elements that act in shorter range.

References associated with point-by-point response:

1. Shan Q, Zeng Z, Xing S, Li F, Hartwig SM, Gullicksrud JA, et al. The transcription factor Runx3 guards cytotoxic CD8(+) effector T cells against deviation towards follicular helper T cell lineage. *Nat Immunol.* 2017;18(8):931-9.
2. Milner JJ, Toma C, Yu B, Zhang K, Omilusik K, Phan AT, et al. Runx3 programs CD8(+) T cell residency in non-lymphoid tissues and tumours. *Nature.* 2017;552(7684):253-7.
3. Zhao DM, Yu S, Zhou X, Haring JS, Held W, Badovinac VP, et al. Constitutive activation of Wnt signaling favors generation of memory CD8 T cells. *J Immunol.* 2010;184(3):1191-9.
4. Jeannot G, Boudousquie C, Gardiol N, Kang J, Huelsken J, and Held W. Essential role of the Wnt pathway effector Tcf-1 for the establishment of functional CD8 T cell memory. *Proc Natl Acad Sci U S A.* 2010;107(21):9777-82.
5. Danilo M, Chennupati V, Silva JG, Siegert S, and Held W. Suppression of Tcf1 by Inflammatory Cytokines Facilitates Effector CD8 T Cell Differentiation. *Cell Rep.* 2018;22(8):2107-17.
6. Gautam S, Fioravanti J, Zhu W, Le Gall JB, Brohawn P, Lacey NE, et al. The transcription factor c-Myb regulates CD8(+) T cell stemness and antitumor immunity. *Nat Immunol.* 2019;20(3):337-49.
7. Schubert LA, Jeffery E, Zhang Y, Ramsdell F, and Ziegler SF. Scurfin (FOXP3) acts as a repressor of transcription and regulates T cell activation. *J Biol Chem.* 2001;276(40):37672-9.
8. Xu Z, Xing S, Shan Q, Gullicksrud JA, Bair TB, Du Y, et al. Cutting Edge: beta-Catenin-Interacting Tcf1 Isoforms Are Essential for Thymocyte Survival but Dispensable for Thymic Maturation Transitions. *J Immunol.* 2017;198(9):3404-9.
9. Johanson TM, Chan WF, Keenan CR, and Allan RS. Genome organization in immune cells: unique challenges. *Nat Rev Immunol.* 2019;19(7):448-56.
10. Johanson TM, Lun ATL, Coughlan HD, Tan T, Smyth GK, Nutt SL, et al. Transcription-factor-mediated supervision of global genome architecture maintains B cell identity. *Nat Immunol.* 2018;19(11):1257-64.
11. Durand NC, Shamim MS, Machol I, Rao SS, Huntley MH, Lander ES, et al. Juicer Provides a One-Click System for Analyzing Loop-Resolution Hi-C Experiments. *Cell Syst.* 2016;3(1):95-8.
12. Rao SS, Huntley MH, Durand NC, Stamenova EK, Bochkov ID, Robinson JT, et al. A 3D map of the human genome at kilobase resolution reveals principles of chromatin looping. *Cell.* 2014;159(7):1665-80.

REVIEWERS' COMMENTS

Reviewer #1 (Remarks to the Author):

The authors have thoroughly addressed the prior critiques. This is a very thoughtful and complete study.

Reviewer #2 (Remarks to the Author):

The authors have made an extremely thorough response to the three reviewers and have further strengthened a paper that was already of great interest. They have addressed each of the issues that concerned me and have added substantial biological impact with the new data they introduced in response to the other reviewers.

Reviewer #3 (Remarks to the Author):

The authors have nicely addressed all of my concerns.

Point-by-point response to REVIEWER COMMENTS

Reviewer #1 (Remarks to the Author):

The authors have thoroughly addressed the prior critiques. This is a very thoughtful and complete study.

Reviewer #2 (Remarks to the Author):

The authors have made an extremely thorough response to the three reviewers and have further strengthened a paper that was already of great interest. They have addressed each of the issues that concerned me and have added substantial biological impact with the new data they introduced in response to the other reviewers.

Reviewer #3 (Remarks to the Author):

The authors have nicely addressed all of my concerns.

We are delighted that all three reviewers are satisfied with our revision. We appreciate their constructive and insightful comments, which greatly improved the manuscript in many aspects. We are excited to share our progress with the research community from the platform of *Nature Communications*.